# Bhalin, an Essential Cytoskeleton-Associated Protein of *Trypanosoma brucei* Linking *Tb*BILBO1 of the Flagellar Pocket Collar with the Hook Complex

**DOI:** 10.3390/microorganisms9112334

**Published:** 2021-11-11

**Authors:** Christine E. Broster Reix, Célia Florimond, Anne Cayrel, Amélie Mailhé, Corentin Agnero-Rigot, Nicolas Landrein, Denis Dacheux, Katharina Havlicek, Mélanie Bonhivers, Brooke Morriswood, Derrick R. Robinson

**Affiliations:** 1Protist Parasite Cytoskeleton (ProParaCyto) Group, CNRS UMR 5234, Fundamental Microbiology and Pathogenicity, University of Bordeaux, 146 Rue Léo Saignat, 33076 Bordeaux, France; christine.reix@gmail.com (C.E.B.R.); celia.florimond@gmail.com (C.F.); anne.cayrel@u-bordeaux.fr (A.C.); amelie.mailhedu12@gmail.com (A.M.); corentinagnerorigot@gmail.com (C.A.-R.); nicolas.landrein@u-bordeaux.fr (N.L.); denis.dacheux@bordeaux-inp.fr (D.D.); melanie.bonhivers@u-bordeaux.fr (M.B.); 2Laboratory of Parasitology, National Reference Center for Malaria, WHO Collaborative Center for Surveillance of Antimalarial Drug Resistance, Pasteur Institute of French Guiana, 97306 Cayenne, French Guiana; 3Société Fromagère de Saint Affrique, Camaras, 12400 Saint-Affrique, France; 4Enstbb, École Nationale Supérieure de Technologie des Biomolécules de Bordeaux, 146 Rue Léo Saignat, 33076 Bordeaux, France; 5Max Perutz Labs, Vienna BioCenter, Dr. Bohr-Gasse 9, 1030 Vienna, Austria; kh.anoniem@gmail.com; 6Department of Cell and Developmental Biology, Biocenter, University of Würzburg, Am Hubland, D-97074 Würzburg, Germany; brooke.morriswood@uni-wuerzburg.de

**Keywords:** trypanosoma, flagellar pocket, hook complex, endocytosis, cytoskeleton, protozoan, flagellar pocket collar

## Abstract

Background: In most trypanosomes, endo and exocytosis only occur at a unique organelle called the flagellar pocket (FP) and the flagellum exits the cell via the FP. Investigations of essential cytoskeleton-associated structures located at this site have revealed a number of essential proteins. The protein *Tb*BILBO1 is located at the neck of the FP in a structure called the flagellar pocket collar (FPC) and is essential for biogenesis of the FPC and parasite survival. *Tb*MORN1 is a protein that is present on a closely linked structure called the hook complex (HC) and is located anterior to and overlapping the collar. *Tb*MORN1 is essential in the bloodstream form of *T. brucei.* We now describe the location and function of BHALIN, an essential, new FPC-HC protein. Methodology/Principal Findings: Here, we show that a newly characterised protein, BHALIN (BILBO1 Hook Associated LINker protein), is localised to both the FPC and HC and has a *Tb*BILBO1 binding domain, which was confirmed in vitro. Knockdown of *BHALIN* by RNAi in the bloodstream form parasites led to cell death, indicating an essential role in cell viability. Conclusions/Significance: Our results demonstrate the essential role of a newly characterised hook complex protein, BHALIN, that influences flagellar pocket organisation and function in bloodstream form *T. brucei* parasites.

## 1. Introduction

*Trypanosoma brucei brucei* is a pathogen, transmitted by the bite of an infected tsetse fly. The subspecies *T. brucei gambiense* and *T. b. rhodesiense* cause the neglected tropical disease Human African Trypanosomiasis (HAT), in West/Central and East/Southern Africa, respectively. *T. b. brucei* contributes to the economically devastating animal wasting disease Animal African Trypanosomiasis (AAT), or nagana, across Africa [1,2]. The disease is invariably fatal if left untreated, and despite major steps towards the elimination of HAT as a public health issue, AAT remains a threat to animal health and human livelihoods [3].

Trypanosomes have evolved mechanisms to survive in the mammalian host, including antigenic variation by periodically changing the surface glycosylphosphatidylinositol-anchored variant surface glycoproteins (VSG). In this respect, the whole surface coat is recycled entirely by endocytosis (solely clathrin dependant) and carried out exclusively at the flagellar pocket (FP), an infolding of the pellicular membrane [4,5,6,7,8]. RNA interference knockdown of the clathrin heavy chain, *TbCLH*, is lethal in both the insect procyclic form (PCF) and the mammalian bloodstream forms (BSF). Furthermore, a grossly enlarged flagellar pocket was observed in BSF after RNAi of *TbCLH* due to a severe imbalance in endocytosis and exocytosis; this phenotype was named “BigEye” [5]. Interestingly, a differential role for actin during the life cycle of *T. brucei* was observed. RNAi knockdown of actin in PCF had no effect whilst it was lethal in BSF leading to a similar BigEye phenotype [9]. Intimately linked to the FP is a cytoskeletal component called the flagellar pocket collar (FPC). The FPC is a cytoskeletal structure that encircles the flagellum at the site where it exits the cell body and forms a boundary or interface at the intersection of the pellicular, flagellar and flagellar pocket membranes [8,10,11,12]. To date, a handful of FP- and FPC-associated proteins have been identified but only one essential protein FPC protein has been characterised, *Tb*BILBO1 [11]. *Tb*BILBO1 is a multi-domain protein with an N-terminal domain (NTD) involved in the interaction with FPC4, two EF-hand calcium-binding motifs (EF), a long coiled-coil domain (CC) involved in protein dimerization, and a leucine zipper domain (LZ) required for polymerisation [13,14,15,16]. RNAi knockdown of *Tb*BILBO1 in PCF cells resulted in aberrant flagellum re-positioning and new flagellum detached from the length of the cell body. Importantly, the detached flagellum did not possess a flagellar pocket, and in these detached flagella, the transition zone, collarette and FP receptors that are normally sequestered within the pocket were exposed to the exterior of the cell. These data demonstrated that *Tb*BILBO1 is necessary for the biogenesis of a new FPC and FP and indeed for cell viability [11].

The FPC is intimately positioned proximal and distal to a cytoskeletal structure called the hook complex (HC). The HC was initially described as a bilobe consisting of a hook structure and a parallel arm [17]. *Tb*Centrin2, *Tb*Centrin4 [18] and *Tb*CAAP (Centrin Arm Associated Protein 1) [19,20] have been localised to the arm structure. Knockdown of *Tb*Centrin2 or *Tb*Centrin4 prevents Flagellum Attachment Zone (FAZ) assembly and affects flagellum attachment. Proteins localised to the hook structure currently are: *Tb*Smee1 [21], *Tb*LRRP1 [22], *Tb*FPC4 [13] and *Tb*MORN1 (Membrane Occupation and Recognition Nexus) [23]. Knockdown of *Tb*Smee1 disrupts the hook morphology and positioning of *Tb*PLK and affects the rate of flagellar pocket macromolecule uptake in PCF [24]. *Tb*LRRP1 was found to be essential in the PCF and knockdown prevented duplication of the centrin arm. *Tb*FPC4 is a microtubule-binding protein and a *Tb*BILBO1 partner linking the FPC and the HC. *Tb*MORN1 is essential in the BSF and protein depletion led to impaired macromolecule internalisation and parasite cell death [25]. In *Tb*MORN1 knockdown, trypanosome death was accompanied by a “BigEye” phenotype caused by enlargement of the flagellar pocket. Uptake assays using fluorescent markers demonstrated that *Tb*MORN1 functions to influence the entry of molecules into the flagellar pocket, suggesting a novel link between the cytoskeleton and the endomembrane system. Recently, two proteins BOH1 (Bait of Hook 1) and BOH2 were described as localising partly to the hook and extending distally between the hook and arm, and whose knockdown impaired flagellum inheritance and proper assembly of the hook complex [20,26].

In the work presented here, we describe BHALIN-BILBO1 Hook Associated Linker protein (Tb927.4.3120) a protein simultaneously discovered during a yeast 2-hybrid (Y2H) analysis using the full genome of *T. brucei* 927 as prey with *Tb*BILBO1 as bait, and a proximity-dependent biotin identification using *Tb*MORN1 as a probe [19]. This suggests that BHALIN is a *Tb*BILBO1 binding protein in very close proximity to *Tb*MORN1 with a strategic location important for parasite viability. Here, we demonstrate that BHALIN interacts with *Tb*BILBO1 and localises to the part of the hook complex that overlaps with the FPC and that knockdown in bloodstream form cells leads to a dramatically distorted, lethal (“BigEye”) phenotype.

## 2. Methods

### 2.1. Cell Lines, Culture and Transfection

Procyclic *Trypanosoma brucei* EATRO1125, procyclic and bloodstream form trypanosomes Tb427 Single marker Oxford strain (SmOx) [27] cell lines, expressing the T7 RNA polymerase and tetracycline repressor, were cultured with selection antibiotic puromycin 1 μg/mL for PCF and 0.1 μg/mL for BSF. Transfectants of SmOx cell line with the RNAi vector, p2T7 Ti^TAblue^ [28] were selected with hygromycin, 25 μg/mL for PCF and 5 μg/mL for BSF. Endogenous tagging of BHALIN (Tb927.4.3120) with 10xTy1 [29] was selected with blasticidin (10 μg/mL for PCF and 5–10 μg/mL for BSF: 5 μg/mL for culture and 10 μg/mL for selection). PCF were grown in SDM-79 medium (GE Healthcare, Chicago, IL, USA, G3344-3005), made up in the laboratory with Aguettant water (OTEC H_2_O, Aguettant Essential Medicine, Lyon, France, 600500); the pH adjusted to 7.4, and completed by the additions of 10% Foetal Bovine Serum (FBS; Gibco, Waltham, MA, USA, 11573397; complement deactivated at 56 °C for 30 min) and 2 mg/mL haemin (Sigma Aldrich, Burlington, MA, USA, H-5533). PCF were incubated at 27 °C. BSF were grown in Iscove’s Modified Dulbecco’s Medium [30], containing: IMDM (Gibco, Waltham, MA, USA, 42200-014); 3.024 g/L Sodium Bicarbonate; 0.136 g/L Hypoxanthine; 0.11 g/L Sodium Pyruvate; 0.039 g/L Thymidine; 0.028 g/L Bathocuproinedisulfonic acid; 2 μM mercaptoethanol; 1.7 μM L-Cysteine; 55 μg/mL Kanamycin; 10% FBS, made up to 1 L with Aguettant water. BSF cells were incubated at 37 °C with 5% CO_2_.

Trypanosomes were transfected with either linearised plasmid or purified PCR products as described in [30,31] before resuspending them in transfection buffer as described in [32]. Transfection cuvettes with a 2 mm gap (VWR International, Radnor, PA, USA; 732-1136) were used in a Biosystems Nucleofector R II, AMAXA, using programme X-001 or Z-001. Clones were obtained post-transfection by serial dilution in 96-well plates, in media containing the selection antibiotic(s) and maintained in logarithmic phase growth at 2 × 10^6^ cells/mL for PCF and 1 × 10^5^ cells/mL for BSF.

U-2 OS cells. Human bone osteosarcoma epithelial cells, ATCC Number: HTB-96 [33] were grown in D-MEM Glutamax (Gibco) supplemented with 10% foetal calf serum and 1% penicillin-streptomycin at 37 °C plus 5% CO_2_. Exponentially growing cells were transfected with 0.5–1 μg DNA using Lipofectamine 2000 in OPTIMEM (Invitrogen, Waltham, MA, USA) according to the manufacturer’s instructions and processed by IFA 24 h post-transfection.

### 2.2. Anti-BHALIN Antibody Production

Six histidine-tagged BHALIN was produced in *E. coli* and purified as described previously for *Tb*FPC4 [13] and used to immunise hens (Eurogentec, Seraing, Liège, Belgium).

### 2.3. Reporter Uptake Assays

The protocol was adapted from [20]. BSF trypanosomes were diluted to 1 × 10^5^ cells/mL and grown overnight at 37 °C and induced with 1 μg/mL tetracycline. A total of 2 × 10^6^ cells from each condition were collected, centrifuged at 800× *g* for 10 min at 4 °C, then resuspended in 1 mL of ice-cold serum-free IMDM. The cells were then pelleted by centrifugation at 800× *g* for 5 min at 4 °C, then resuspended in 100 μL of ice-cold serum-free IMDM, and kept on ice for 10 min. Dextran-fluorescein (10,000 MW, anionic, Life Technologies Corporation; ref. D1821) final concentration 5 mg/mL and Texas Red-conjugated Concanavalin A (Life Technologies Corporation; ref. C825) to a final concentration of 10 μg/mL, were added to trypanosomes in tubes and mixed by flicking. The cells were then incubated for 15 min on ice. Half the samples were fixed (time, t = 0), the other incubated at 37 °C for 30 min (t = 30 min). For both sets of samples, uptake was stopped by the addition of 1 mL of ice-cold serum-free IMDM and cells were immediately fixed for immunofluorescence.

### 2.4. Immunofluorescence Assay

Trypanosoma: Trypanosomes were prepared and fixed as described in [14] and deposited on poly-L-lysine 0.1% solution (Sigma-Aldrich, Burlington, MA, USA; P8920) coated slides (Thermo Scientific, Waltham, MA, USA, Cel-line diagnostic microscope slides, 30-225H-RED-CE24) for 4 min to adhere. Whole cells were fixed in methanol at −20 °C for at least 60 min. Detergent extraction was with 0.25% Nonidet P-40 (IGEPAL) in 100 mM PIPES-NaOH pH 6.8 (Piperazine-N,N’-bis-(ethanesulfonic acid; Euromedex; 1124), 1 mM MgCl_2_ for 5 min. Bloodstream form cells were washed in vPBS (NaCl 0.8 mg/mL, KCl 0.22 mg/mL, Na_2_HPO_4_ 22.7 g/mL, KH_2_PO_4_ 4.4 mg/mL, sucrose 15.7 mg/mL, glucose 1.8 mg/mL) and fixed on Thermo Scientific Superfrost^R^ plus 1800AMNZ slides. After two PBS washes for 5 min, fixed trypanosomes were incubated with primary antibodies in PBS for 1 h in a dark moist chamber (anti-BHALIN hen 1:500; anti-*Tb*BILBO1 1-110 rabbit [14] 1:4000 dilution; anti-Ty1 BB2 mouse [34] 1:200; anti-*Tb*MORN1 [23] rabbit 1:5000; anti-PFR2 rabbit 1:200; anti-cMyc 9E10 mouse [35] 1:100; anti-cMyc rabbit (Sigma-Aldrich, Burlington, MA, USA, C3956) 1:200; anti-FTZC rabbit 1:10,000) [36]. Following the primary antibody incubation, samples were washed twice (5 min) in PBS and incubated 1 h with the fluorophore-conjugated secondary antibodies: Alexa594-cojugated anti-chicken (Molecular Probes A11042), FITC-conjugated goat anti-mouse (Sigma F2012), Alexa594-conjugated goat anti-rabbit (Thermo fischer, Waltham, MA, USA, A11012). After two 5 min washes in PBS, DNA was labelled for 2 min with DAPI (10 μg/mL) followed by two PBS washes. Slides were mounted with SlowFade Gold (Thermofischer S-36936).

For fixation of BSF from endocytotic assays, after incubation with fluorophore-conjugated dextran and ConA, cells were pelleted at 800× *g* for 5 min at 4 °C and then resuspended and incubated in 1 mL 4% paraformaldehyde–0.1% glutaraldehyde (Sigma G5882) in ice-cold vPBS for 20 min on ice. Cells were centrifuged at 800× *g* for 5 min at 4 °C, resuspended in 50 μL vPBS at 4 °C and 50 μL was spread over a slide well coated in poly-L-lysine. Cells were left to adhere for 20 min at room temperature (RT), then dried for 10 min. Cells were permeabilised with 0.25% Triton TX-100 in PBS for 4 min, washed twice for 5 min with 100 mM glycine, washed twice for 5 min with PBS, incubated with 25 μL DAPI (Sigma D-9542) for 2 min at RT, washed twice with PBS, then a drop of Slowfade Gold anti-fade reagent (Thermofischer S-36936) was added to each well before covering with a coverslip (Knittel Glass, Braunschweig, Germany) and sealing with nail varnish. Trypanosomes were imaged directly, using the same acquisition settings and exposure times for both non-induced and induced samples.

U2-OS cells: U-2 OS cells grown on glass coverslips were washed with PBS, fixed in 3% paraformaldehyde for 15 min (at 37 °C) and permeabilised 30 min in PBS containing 10% FBS and 0.1% saponin or briefly extracted with an extraction buffer (0.5% TX-100, 10% glycerol in EMT (60 mM PIPES-NaOH pH6.9, 25 mM HEPES, 10 mM EGTA, 10 mM MgCl_2_) and fixed in 3% paraformaldehyde in PBS for 15 min (at 37 °C). Samples were processed for immunofluorescence as in [14]. The primary antibodies (anti-*Tb*BILBO1 1–110 rabbit [14] 1:4000 dilution; anti-GFP living colours rabbit (Clontech, Fitchburg, WI, USA. 632460) 1:1000 dilution; anti-Ty1 mouse BB2 [34] 1:200 dilution; anti-*Tb*MORN1 [23] 1:5000 dilution) were incubated for 1 h in a dark moist chamber. After two PBS washes, cells were incubated for 1 h with the secondary antibodies anti-rabbit IgG conjugated to Alexa fluor 594 (Molecular Probes, 1:400); anti-mouse IgG conjugated to FITC (Sigma, 1:400). The nuclei were stained with DAPI (0.20 μg/mL in PBS for 5 min), then washed twice in PBS and mounted overnight with Prolong (Molecular Probes S-36930). Images were acquired on a Zeiss Imager Z1 microscope with Zeiss 100x or 63x objectives (NA 1.4), using a Photometrics Coolsnap HQ2 camera and Metamorph software (Molecular Devices, San Jose, CA, USA), and processed with ImageJ.

Transmission electron microscopy (TEM): 10 mL of mid-log phase bloodstream form cells (WT and RNAi induced for 24 h) were fixed, dehydrated and embedded as previously described in [37].

Immuno-electron microscopy (iEM) of trypanosomes: 10 mL of mid log phase 5 × 10^6^ /mL PCF cells were harvested (1000× *g* for 5 min), washed twice with PBS by centrifugation (1000× *g* for 5 min) then re-suspended in 500 μL of PBS. Freshly glow discharged, butvar and carbon-coated G2000-ni nickel grids (EMS) were floated on the droplets and the cells were allowed to adhere for 15 min. Grids were then moved onto a drop of 1% NP-40 in PEME buffer (100 mM PIPES-NaOH pH 6.8, 1 mM MgCl_2_, 0.1 mM EGTA) for 5 min at RT, followed by incubation on a 500 μL drop of 1% NP-40, 1 M KCl in PEME buffer to depolymerise the sub-pellicular microtubules for 3 × 5 min, 4 °C). Flagella were then washed 2 × 5 min in PEME buffer at RT, blocked by transferring to four 50 μL drops of 2% fish skin gelatin (Sigma-Aldrich, Burlington, MA, USA, G7041) or 0.5% BSA, 0.1% tween 20 in PBS and then incubated in 25 μL of primary antibody diluted in blocking buffer (1% fish skin gelatin in PBS). Each primary antibody was used either alone, or mixed with a second primary antibody for double labelling, for 1 h at RT: anti-Ty1 BB2 mouse [34] 1:5 dilution, anti-*Tb*MORN1 [23] 1:400 dilution; anti-*Tb*BILBO1 1-110 rabbit [14] 1:200 dilution. The grids were then moved through four drops of blocking solution and incubated in a secondary antibody for 1 h at RT (anti-mouse goat GMTA 5 nm gold, and/or anti-rabbit goat 15 nm gold GAR15, BBI solutions, Crumlin, Wales, UK). Grids were then incubated in blocking solution 4 × 5 min at RT, then 2 × 5 min in 0.2% fish skin gelatin in PBS and fixed in 2.5% glutaraldehyde in 0.2% fish skin gelatin in PBS. Flagella were negatively stained with 1% aurothioglucose, 10 μL for 30 s (Merck, Kenilworth, NJ, USA, 1045508). Micrographs were taken on a Phillips Technai 12 transmission electron microscope at 100 kV.

### 2.5. Molecular Biology and Cloning

Endogenous tagging was carried out as described in [29]. To generate cell lines expressing endogenously 10xTy1 tagged BHALIN (_Ty1_BHAlin, BHALIN_Ty1_) PCR was performed with the pPOTv7-BLAST-10xTy1 vector template; the PCR product was transfected as described in [29]. For BSF transfections a nucleic acid purification step was carried out prior to the ethanol precipitation of PCR product for transfection. Verification that the tag had been inserted in the trypanosomes was made by PCR on genomic DNA extracted from transfected cell lines and sequencing.

Cloning of *BHALIN* recoded gene: full-length and truncations for replacement of one allele in *T. brucei.* A recoded RNAi-resistant nucleotide sequence for *BHALIN* (Tb947.4.3120) was produced (cloned in pEX-K4, Eurofins Genomics, Ebersberg bei München, Germany). To produce a construct for the full-length 10xcMyc N-terminal tagged BHALIN protein (_cMyc_BHALIN^rec^), the full-length *BHALIN* recoded gene was amplified by PCR and cloned between the *BamH*I-*Sac*I pPOTv7-NEO-10xcmyc vector sites. A final PCR was carried out using primers containing 80 bp homology with the 5′ and 3′ UTR for the gene *BHALIN*, to replace one allele of *BHALIN* with the full-length *_cMyc_BHALIN^rec^* sequence and inserting an antibiotic cassette for selection (neomycin). The truncations of *BHALIN* recoded gene *_cMyc_BHALIN^rec^-T1* (corresponding to aa 1-595), *_cMyc_BHALIN^rec^-T2* (aa 596-800) and *_cMyc_BHALIN^rec^-T3* (aa 596-933) were obtained by overlapping PCRs using the vector pPOTv7-NEO-10xcMyc as a template and the PCR product of the desired sequence of *BHALIN* recoded gene amplified from pEX-K4 plasmid with an overhang homologous to the pPOTv7 vector at the 5′. In the second step, the two PCR products from the first step were mixed and a PCR reaction run using primers designed with overhangs of 80 bp matching the 5′ and 3′ UTR of *BHALIN*, to allow for the replacement of one allele of *BHALIN*. The final PCR products were purified by ethanol precipitation before transfection in trypanosomes.

### 2.6. Cloning Full-Length BHALIN and Truncations for Transfection into U-2 OS Cells

The in-house made vector pcDNA3.1-3Ty1-X was used to clone the full-length and truncations of *BHALIN* sequence. The sequences were amplified by PCR and were cloned between the *BamH*I and *Xba*I sites using AQUA cloning [38]. The sequences (vectors and endogenous tagging) were controlled by DNA sequencing (Eurofins, Ebersberg bei München, Germany).

### 2.7. Extraction of RNA

Starting with 1 × 10^8^ PCF cells or 2 × 10^7^ BSF cells, trypanosomes were centrifuged at 800× *g* for 10 min at 4 °C, washed once in PBS (vPBS for BSF) at RT. The pellet was resuspended in 0.5 mL of TRIzol (PCF) or 0.400 mL Isol-RNA Lysis Reagent (5 PRIME 2302700) (BSF) following the manufacturer’s instructions. The tube was incubated for 5 min at RT, then 0.1 mL of chloroform was added. The tube was mixed vigorously (by vortexing) for 15 s, before centrifugation at 4 °C for 20 min at 12,000× *g*. The RNA remaining in the upper aqueous phase was recovered and was precipitated by mixing with 250 μL of isopropyl alcohol and incubated for 1 h, at −80 °C. The tube was then centrifuged at 4 °C for 10 min at 16,000× *g* and the supernatant discarded. An amount of 0.5 mL of 100% isopropanol (or 70% ethanol) was added to the RNA pellet (1 mL for 1 mL of TRIzol) and the tubes were stored at −20 °C. When ready for use, the tubes were centrifuged at 7500× *g* for 5 min at 4 °C, and the pellet is re-suspended in 50 μL ddH_2_O. Prior to RT-PCR, the extracted RNA samples were subjected to treatment with Turbo DNA-free TM (Invitrogen, Waltham, MA, USA, AM1907), according to the manufacturer’s instructions.

### 2.8. Semi-Quantitative RT-PCR

The purified RNA was used as a template for PCR, with 25 programme cycles to allow a semi-quantitative measurement of mRNA. Primers for RT (reverse transcriptase)-PCR were designed to be specific for either *BHALIN* wild-type sequence or *BHALIN* recoded sequence. Positive and loading controls were used: 18 s ribosomal unit, TERT (telomerase reverse transcriptase) and Hsp60 (heat shock protein 60). SuperScript III One-step RT-PCR with Platinum Taq (Invitrogen, Waltham, MA, USA, 12574-018) kit was used and the reaction performed according to the manufacturer’s instructions. Measurements of the amount of amplified nucleic acid obtained were made using G:BOX (Syngene, Cambridge, UK).

### 2.9. Western Blot Analysis

Cells were collected, centrifuged at 800× *g* for 10 min, and washed in PBS for PCF, or vPBS for BSF. For whole cells, the sample was directly re-suspended in protein sample buffer 2x plus Benzonase^®^ nuclease (Sigma, Ref. E1014). For detergent-extracted cells (cytoskeletons), the washed cells were re-suspended in 1% NP40 (Igepal), 100 mM PIPES, 1 mM MgCl_2_ for 7 min; cytoskeletons were checked by microscopy, then washed in 100 mM PIPES buffer before resuspension in sample buffer 2x. The samples were boiled at 100 °C for 5 min then stored at −20 °C until required. SDS-PAGE gels were prepared and samples loaded at 5 × 10^6^ cells-equivalent per well; 8% SDS-PAGE was used. Samples were transferred in a semi-dry system (Bio-Rad Trans-Blot^®^ Semi-dry transfer cell 221BR54560) onto Polyvinylidene difluoride (PVDF) membrane and blocked with 5% milk, 0.2% Tween 20 in TBS (blocking solution; BS) for 60 min. Primary antibodies were diluted in BS and incubated with the membranes overnight at 4 °C: anti-Ty1 mouse [34] 1:50,000, anti-cMyc rabbit 1:1000 (Sigma, Ref. C3956), anti-*Tb*BILBO1 1-110 rabbit 1:1000 [13], anti-*Tb*MORN1 rabbit 1:20,000 [23], anti-*Tb*SAXO mouse [39], anti-PFR2 rabbit 1:2000 (Baltz/Biteau), anti-Enolase [40] 1:25,000 and anti-tubulin TAT1 mouse 1:1000 [41] were used as loading controls. Membranes were washed in TBS and then BS before probing with a secondary antibody conjugated to Horse Radish Peroxidase (HRP) diluted 1:10,000 in BS anti-mouse (Jackson, sheep, 515-035-062) or anti-rabbit (Sigma, goat, A9169) for 60 min at RT. Visualisation was made using Image Quant LAS 4000 (luminescent Image Analyser) or Chemidoc (Bio-Rad, Hercules, CA, USA).

### 2.10. Yeast Two-Hybrid Assay

Interactions assays were performed on SC-W-L-H medium as described in the online protocol (dx.doi.org/10.17504/protocols.io.btzenp3e). Photos were acquired after 3 days of incubation.

## 3. Results

### 3.1. BHALIN Is a Trypanosomatid Specific Protein

Bioinformatic analysis of the *BHALIN* gene (Tb927.4.3120) using BLASTn [42] detected orthologues only in *Trypanosoma* spp as previously described [43]. Over 99% sequence identity was observed between *T. b. brucei*, *T. b. gambiense* and *T. equiperdum* indicating a close similarity between these sub-species [44,45,46,47]. A phylogenetic tree constructed from the syntenic orthologues of *BHALIN* [48] further emphasised the high level of sequence conservation (Figure 1A). Apart from a globular domain from aa 596-759, in silico analysis was unable to predict any protein functional domains (Figure 1B). The original Y2H *Tb*BILBO1 genomic screen identified aa 661-781 within the BHALIN protein as the BI LBO1-binding domain (B1B) which is mostly part of the globular domain. We used yeast two-hybrid assays (Y2H) to confirm the binding of BHALIN to *Tb*BILBO1 (Figure 1C). Indeed, full-length BHALIN interacted with full-length *Tb*BILBO1. Further, BHALIN interacted with the *Tb*BILBO1 truncations T2 and T3 that both bear the EF-hand domains (EF), but not with truncation T4 devoid of the EF-hand domain. This suggests that BHALIN B1B interacts with the *Tb*BILBO1 EF-hands domain.

### 3.2. BHALIN Forms a Hook-Shaped Structure at the Hook Complex

BHALIN was in situ epitope-tagged with 10xTy1 tag on either the N- or C-termini (see Methods [29]). Immunofluorescence assays on PCF detergent-extracted parasites (cytoskeletons) revealed that the location of the tagged protein was similar regardless of whether C-terminal (Figure 2A) or N-terminal (Figure 2B–D) tagging was used. The BHALIN signal was observed at the FPC/hook complex region, forming a distinctive hook-shaped structure and indicating that it was cytoskeleton-associated. Endogenous 10xTy1 tagging of BHALIN had no effect on growth rate of either PCF or BSF as compared to the wild-type parental cells (WT) (for PCF in Appendix A and BSF in Appendix A). All further experiments were carried out using the N-terminal 10xTy1 tag. (D) The growth curve of BSF WT and RNAi BHALIN non-induced (-tet) and induced (+tet) was done over a time period of 48 h.

To determine the location of BHALIN in relation to other known FPC and hook complex proteins, immunofluorescence assays were performed with antibodies to important and characterised cytoskeletal structures. *Tb*BILBO1 is located on the FPC [13], and we observed that BHALIN partially co-localises with *Tb*BILBO1 at the FPC, in PCF (Figure 2A) and BSF (Figure 2B). *Tb*MORN1 is a marker for the hook complex, forming a distinctive hook-shaped structure [23] and BHALIN co-localised with *Tb*MORN1 with a strongest labelling at the head of the hook (Figure 2C). To determine the location of BHALIN in relation to the flagellum, a monoclonal antibody against the paraflagellar rod (PFR) [49] was used. In Figure 2D, the hook-shaped structure labelled by 10xTy1-tagged BHALIN can be seen at the base of the flagellum and then passing a short way distally and parallel to the PFR. Immuno-electron microscopy was used to explore in more detail the precise location of BHALIN, using 5 nm gold beads against the 10xTy1 endogenous tag. The gold beads were observed at the FPC curving around the flagellum, then extending distally along the flagellum, gradually diminishing and tapering off (Figure 2E). Additionally, BHALIN co-localises with *Tb*MORN1 (Appendix A). In summary, BHALIN localisation overlaps with *Tb*MORN1 and *Tb*BILBO1 and is clearly in close proximity to both proteins as shown by our IFA and iEM studies supporting the identification of BHALIN in the *Tb*MORN1 proximity-dependent biotin experiment [19].

### 3.3. BHALIN Is Present at the Hook Complex throughout the Cell Cycle

*T. brucei* progresses through a clearly determined sequence of cell cycle events [50]. In summary: G1 phase cells have a single kinetoplast (K, the mitochondrion genome), and a single nucleus (N), i.e., 1K1N. The kinetoplast is the first to divide, along with a new flagellum and flagellar pocket to produce a 2K1N cell; mitosis is next to produce a 2K2N cell. In both PCF and BSF, the main phases are the same, but the linear order of the kinetoplast and nucleus (posterior to anterior) is different. Specifically, the 2K2N stage in PCF is KNKN but in BSF it is KKNN. In Figure 2F (PCF) and 2G (BSF), the main phases of the cell cycle are illustrated. Labelling of BHALIN was observed in PCF and BSF trypanosomes through the cell cycle (Figure 2F,G). In Figure 2G, the cells are also co-labelled with MORN1. BHALIN often appear to have a stronger signal and be most hook-like in the 1K1N stage (Figure 2(Fa)). As the new flagellum grew and a new FP was formed, the BHALIN signal could be observed as two structures of equivalent size and intensity (Figure 2(Fb,c). A close analysis of the immunolabelling in the BSF show that BHALIN signal is sometimes vestigial on the old HC, (Figure 2(Gb,d)). As the kinetoplast divided, followed by mitosis (Figure 2F,(Gd)), the signal for BHALIN remained present at both hook complexes (Figure 2F,(Ge)).

### 3.4. Binding of BHALIN Modifies the Shape of the TbBILBO1 Filaments and Requires the TbBILBO1 EF-Hand Domain

Orthologues for BHALIN were not identified in organisms other than trypanosomatids; however, expressing trypanosome-specific proteins in a novel environment can give insightful information about the properties of the protein. Indeed, the ectopic expression in mammalian U-2 OS cells allows for the assay for self-assembly under conditions of cytosolic crowding, which are difficult to replicate in vitro. Additionally, we can screen for association with cellular structures (such as the microtubule network). Previously published data demonstrated that when *Tb*BILBO1 is expressed in U-2 OS cells it forms linear polymers with globular or comma-shaped ends (termini) [14] (Figure 3(Aa)). When _Ty1_BHALIN was expressed in U-2 OS cells, a cytoplasmic labelling was observed, suggesting that BHALIN does not form polymers and does not associate with mammalian cytoskeletal structures (Figure 3(Ab)). When *Tb*BILBO1 and _Ty1_BHALIN were co-expressed, BHALIN co-localised exclusively on the structures formed by *Tb*BILBO1, with the shape and form of the polymers created by *Tb*BILBO1 (Figure 3(Ac)). This indicated that full-length BHALIN binds to *Tb*BILBO1 in absence of other trypanosome-specific proteins. Interestingly, in presence of BHALIN, the *Tb*BILBO1 filaments had a different shape with more globular structures (compare the *Tb*BILBO1-labelled structures in Figure 3(Aa,c)). However, no quantification was performed due to the variation of expression levels of *Tb*BILBO1 and BHALIN from cell to cell. To gain information on the function of individual domains of BHALIN, the Ty1-tagged truncated versions _Ty1_BHALIN T1 (aa1-595), T2 (aa 596-800 flanking the theoretical *Tb*BILBO1 binding domain B1B), and T3 (aa 596-933) were co-expressed with *Tb*BILBO1 (Figure 3(Ad–f)). When _Ty1_BHALIN-T1 was co-expressed with *Tb*BILBO1, the typical *Tb*BILBO1 polymers were formed but _Ty1_BHALIN-T1 was cytoplasmic and seen throughout the entire cell but not on *Tb*BILBO1 (Figure 3Ad), indicating that the N-terminal domain of BHALIN alone is not sufficient to bind to *Tb*BILBO1. However, when _Ty1_BHALIN-T2 was co-expressed with *Tb*BILBO1, identical localisation patterns were observed for both proteins (Figure 3(Ae)), indicating that T2 is required and sufficient to bind to *Tb*BILBO1. A similar co-localisation pattern (but different structures compared to T2) was observed when _Ty1_BHALIN-T3 was co-expressed with *Tb*BILBO1 (Figure 3(Af)); T3 includes T2 but has a longer C-terminal domain of 133 aa. These results confirmed that the *Tb*BILBO1 binding domain is present within the aa sequence 596-800 of BHALIN and that it is required and sufficient to bind to *Tb*BILBO1.

Our Y2H assays suggested that the EF-hand-domain of *Tb*BILBO1 was the interaction domain for BHALIN (Figure 1C). To investigate further the interactions between truncations of *Tb*BILBO1 and BHALIN we used the previously described *Tb*BILBO1 constructs allowing the expression of the polymer-forming *Tb*BILBO1-T3_GFP_ and *Tb*BILBO1-T4_GFP_ [14]. Expression of *Tb*BILBO1-T3_GFP_ produces long filaments on which BHALIN binds to when co-expressed suggesting that the *Tb*BILBO1 NTD is not involved in the interaction. As previously described, *Tb*BILBO1-T4_GFP_ produced spindle-like polymers without noticeable globular ends [14]. When co-expressed with this construct lacking the EF-hand motifs, BHALIN did not localise onto the *Tb*BILBO1-T4_GFP_ polymers and was observed as cytoplasmic (Figure 3(Bd)).

### 3.5. Knockdown of BHALIN in Procyclic Forms Leads to an Altered Cell Cycle Profile

To explore the biological function of BHALIN, inducible knockdown of the protein was employed [32]. RNAi knockdown of BHALIN in PCF trypanosomes did not affect cell growth (Figure 4A); non-induced (NI) and tetracycline-induced cell lines (RNAi) grew at the same rate as wild-type (WT). After RNAi induction, BHALIN protein levels were reduced to an average of 44% at 48 h post induction (hpi) compared with non-induced levels (Figure 4B,C). *Tb*BILBO1 levels were also quantified and, although we saw a trend in protein levels that followed BHALIN levels, the average did not fall below 60% of NI levels. To explore any effects on cell division of BHALIN knockdown on PCF trypanosomes, cell cycle counts were performed, counting 200 cells at each time point of induction (*n* = 3), (Figure 4D). A significant reduction in 1K1N cell cycle stage was observed between WT parasites and those induced for RNAi for 96 h (*p* < 0.05). A slight increase in 2K1N cell cycle stage trypanosomes was observed between WT and 72 hpi (*p* < 0.01) and WT and 96 hpi (*p* < 0.002), student t-test. A slight increase in the percentage of parasites in the 2K2N cell cycle stage was also observed between WT and 48 hpi (*p* = 0.01) and WT and 96 hpi (*p* = 0.02). These data indicate that after RNAi knockdown of BHALIN, PCF trypanosomes either undergo an acceleration of entry from 1K1N to 2K1N and delay from 2K1N to 2K2N and/or delay in cytokinesis, resulting an accumulation of parasites in both 2K1N and 2K2N cell cycle stages.

In cells induced for *Tb*BILBO1 RNAi knockdown, no new FP and no new FPC were formed, the new flagellum-to-cell body attachment was disrupted and the transition zone of the mature basal body was external to the cell body [11]. Interestingly, in *Tb*BILBO1 RNAi knockdown cells, BHALIN labelling was still observed at the FPC of the old flagellum (OF) but was associated with the proximal end of the new flagellum (NF) (Figure 4E, left panel) and localised proximal to the transition zone of the mature basal body (labelled with anti-FTZC a mature basal body marker) of the new flagellum (Figure 4E, right panel). This might indicate that knockdown of *Tb*BILBO1 did not prevent BHALIN expression but in absence of its final target (no new FPC and new HC structures), BHALIN docks onto the BBs.

### 3.6. Knockdown of BHALIN in Bloodstream Forms Is Lethal and Induces a “BigEye” Phenotype

Because *Tb*MORN1 RNAi was lethal in BSF only, we generated a BHALIN RNAi BSF cell line. The induction of BHALIN RNAi knockdown in BSF led to a severe growth defect and cell death within 48 h of induction, with complete population death by 168 hpi (Figure 5A). Non-induced trypanosomes grew similarly to WT (Appendix A). The RNAi specificity was tested by semi-quantitative RT-PCR (Figure 5B) showing a decrease in *BHALIN* mRNA level. This decrease led to a dramatic decrease in BHALIN protein after 24 h of RNAi induction as shown by Western blotting analysis (Figure 5C). Initial observations using light microscopy of live BSF cells after 24 h of BHALIN knockdown revealed a rounding up of the parasites (Figure 5D). To observe the effect of knockdown in more detail, thin sections of RNAi induced parasites were prepared for transmission electron microscopy (Figure 5E). Micrographs of thin sections revealed a “BigEye” phenotype where the flagellar pocket had become enlarged and eventually filled almost the entire trypanosome cell with reorientation of membrane boundaries where, as previously observed by Morriswood and Schmidt [25], the abnormal intracellular location of extracellular structures, such as the axoneme and paraflagellar rod, were observed in these knockdown cells. In conclusion, knockdown of BHALIN by RNAi in BSF trypanosomes led to an enlargement of the FP and rapid parasite death.

### 3.7. Endocytosis Is Perturbed in BSF BHALIN RNAi Cells

To test the hypothesis that the “BigEye” phenotype observed in BSF after RNAi knockdown of BHALIN phenocopies *Tb*MORN1 RNAi knockdown phenotypes, uptake assays were performed (adapted from [25]). In these assays, WT or BHALIN RNAi trypanosomes (induced for 17 h) were incubated with differently sized fluorescent molecules: dextran 10,000 Da (~20 Å diameter) and lectin Concanavalin A (ConA) (~80 Å diameter) [51]. Dextran enters the cell via the fluid phase whilst ConA binds to glycoproteins close to the neck of the pocket and is then taken up by endocytosis. Firstly, WT trypanosomes were incubated on ice for 15 min to prevent endocytosis [52], and fluorescent Dextran and ConA were added for 15 min at 4 °C. At this time point (t = 0), aliquots of cells were fixed and visualised. Labelling could be observed for Dextran and ConA within the FP (Figure 5F). At this stage, not all trypanosomes were labelled with ConA; conversely all cells had Dextran flagellar pocket labelling. Next, the cells were incubated at 37 °C for 30 min to allow endocytosis to resume. The cells were fixed and visualised showing both Dextran and ConA present within the cell body in the endosomal–lysosomal system, indicating that both molecules had been internalised and that the endocytic process was active. At this stage, some ConA could still be observed in the flagellar pocket, whilst all the dextran had been internalised (Figure 5G). After induction of BHALIN RNAi knockdown for 17 h, trypanosomes were also incubated on ice for 15 min followed by the Dextran and ConA incubation for 15 min on ice (time point, t = 0). Dextran was observed inside the now-enlarged flagellar pocket; however, ConA was seen in close proximity to the flagellar pocket and kinetoplast but not co-localising with dextran (Figure 5H). In contrast to WT cells, after warming the induced cells for 30 min at 37 °C the pattern of labelling was unchanged. Dextran was observed filling the enlarged flagellar pocket and ConA remained at a distinct location in the region of the neck of the flagellar pocket (Figure 5I) showing that neither molecule had been taken farther into the cell. This indicated that endocytosis at the FP was impeded in BHALIN RNAi-induced cells producing the same effect as *Tb*MORN1 RNAi knockdown. Interestingly, ConA did not enter the FP but appeared to be located at the edge of the pocket on the external surface of the cell.

### 3.8. Expression of RNAi-Resistant BHALIN Rescues the Lethal Phenotype in BSF

To test the role of different domains of BHALIN in vivo, a recoded version of the *BHALIN* gene was generated (Eurofins Genomics, Germany). This recoded gene provided a nucleotide sequence that was resistant to RNAi but coded for the same amino acids as the native protein (a technique previously shown in *T. brucei* by [53]). This approach was fully successful in BSF to generate a _Ty1_BHALIN RNAi-inducible cell line with the second allele replace with a recoded 10xcMyc-tag BHALIN gene (_cMyc_BHALIN^Rec^). In the _Ty1_BHALIN RNAi-inducible PCF cell line, attempts to replace the second allele with a recoded 10xcMyc-tag BHALIN gene in PCF failed, but resulted in an ectopic recoded copy. We used immunofluorescence assays (IFA) to test the location of the recoded full-length protein and confirmed in detergent-extracted cells that _cMyc_BHALIN^Rec^ localises similarly to the WT BHALIN protein both in PCF and BSF (Figure 6B).

Similar to the absence of phenotype after BHALIN RNAi induced PCF cells (Figure 4), BHALIN RNAi induced cells expressing _cMyc_BHALIN^rec^ (which were thereby resistant to RNAi) were not affected. More interestingly, induction of BHALIN RNAi in BSF expressing both _Ty1_BHALIN _and cMyc_BHALIN^rec^ showed no “BigEye” phenotype (Figure 6C). This indicates that _cMyc_BHALIN^rec^ was able to compensate for the loss of the _Ty1_BHALIN protein observed by IFA and WB. Indeed, examination of detergent-extracted cytoskeleton preparations in the 2K1N cell cycle stage by IFA revealed that the fluorescence signals for the _Ty1_BHALIN and the _cMyc_BHALIN^rec^ were present at both the old (black arrow) and new (white arrow) hook complex in non-induced cells (Figure 6Ca). After early RNAi induction _Ty1_BHALIN was only present at the old FPC (black arrow) and not at the new FPC (white arrow) (Figure 6Cb). However, _cMyc_BHALIN^rec^ remained present at both the old and new FPC. Furthermore, induced cells had no growth defect (Figure 6D). Western blotting assays confirmed that the _Ty1_BHALIN protein was considerably reduced at 24 hpi (Figure 6E); however, _cMyc_BHALIN^rec^ remained expressed after 24 hpi RNAi. This confirmed that the recoded protein had not been targeted by RNAi. Quantification of these Western blots by densitometry measurement is shown in Figure 6F (*n* = 3) and illustrated that levels of _Ty1_BHALIN were reduced to less than 20% of non-induced levels at 24 hpi. However, _cMyc_BHALIN^rec^ levels increased to more than 30% above non-induced levels at 24 hpi; both *Tb*BILBO1 and *Tb*MORN1 levels remained unchanged. These data confirmed that the recoded BHALIN mRNA was resistant to RNAi and the expressed recoded protein was functional and rescued the lethal RNAi phenotype.

### 3.9. Deletion of the BILBO1-Binding and C-Terminal Domains Relocates BHALIN to the Cytoplasm

To understand domain function, we attempted to replace, as described above, one WT allele of BHALIN with a recoded T1 BHALIN, truncation (_cMyc_BHALIN^rec^-T1, aa1-595). This approach was unsuccessful in BSF, but we could express it in PCF (Appendix A). In detergent-extracted cytoskeletons (CSK), a weak IFA _cMyc_BHALIN^rec^-T1 labelling was observed near the shank of the hook complex most likely due to binding to pre-existing BHALIN or to BHALIN binding proteins. Interestingly, a faint signal along the FAZ and at the anterior end of the cell body was observed suggesting that BHALIN-T1 domain could play a role in an association with the MTQ-FAZ structure (Appendix A). However, in whole cell preparations (WC), a diffuse cytoplasmic signal was observed, indicating that most of the protein was cytosolic (Appendix A). _cMyc_BHALIN^rec^-T1 was further observed by Western blotting on whole cells (WC) but extremely weak on detergent-extracted cytoskeleton (CK) samples (Appendix A) suggesting that the IFA labelling observed on cytoskeleton might be due to weak binding and or incomplete extraction. After induction of RNAi knockdown of _Ty1_BHALIN, the cytoplasmic _cMyc_BHALIN^rec^-T1 labelling remained. The cytosolic pool of _cMyc_BHALIN^rec^-T1 was further observed by Western blotting on whole cells (WC) (Appendix A). Following induction of RNAi in this _cMyc_BHALIN^rec^-T1 expressing cell line _Ty1_BHALIN expression levels was reduced but the protein was still detected, whilst _cMyc_BHALIN^rec^-T1 and *Tb*BILBO1 levels remained constant and no phenotype was observed. The cytoplasmic localisation of _cMyc_BHALIN^rec^-T1 strongly suggests that the B1B domain is required for localisation at the FPC-HC.

### 3.10. The BHALIN TbBILBO1-Binding Domain (T2) Co-Localises with TbBILBO1 In Vivo

The replacement of one allele of BHALIN with the _cMyc_BHALIN^rec^-T2 (aa 596-800), containing the *Tb*BILBO1-binding domain (B1B) (Figure 1B, Appendix A) was obtained in PCF trypanosomes. It is worth noting that PCR analysis indicated that the recoded _cMyc_BHALIN^rec^-T2 did not replace a native allele but was inserted in an unknown site in the genome; nevertheless, it was still expressed. All attempts to generate a BSF version were unsuccessful. However, IFA showed _Ty1_BHALIN co-localisation with _cMyc_BHALIN^rec^-T2 at the FPC (Appendix A) but with minor differences; the labelling of _Ty1_BHALIN (yellow) extends from the typical hook shape distally along the flagellum (white arrowhead), plus a very weak signal towards the basal bodies (white asterisk), the latter coinciding with the MTQ. The magenta labelling for _cMyc_BHALIN^rec^-T2 was similar to _Ty1_BHALIN but also extended towards the basal bodies (white asterisk), as previously reported for *Tb*BILBO1 suggesting MTQ labelling [13]. Indeed, the location of the _cMyc_BHALIN^rec^-T2 in *T. brucei* appeared almost identical to *Tb*BILBO1 (Appendix A). Importantly, the identification and prediction of this *Tb*BILBO1-binding domain (B1B) was originally made by Y2H assays and could now be confirmed in trypanosomes. Upon induction of RNAi knockdown of _Ty1_BHALIN, the trypanosome cell line expressing _cMyc_BHALIN^rec^-T2 continued to grow similarly to WT (NI) and parental cell lines BHALIN (NI/RNAi) (Appendix A). However, when samples of induced parasites collected at 24 h intervals post RNAi induction were probed by Western blot, a surprising finding was observed; _cMyc_BHALIN^rec^-T2 protein levels diminished after RNAi induction to a greater extent than full-length _Ty1_BHALIN WT protein (Appendix A). This loss of the _cMyc_BHALIN^rec^-T2 protein after induction of RNAi knockdown was an unexpected finding because the recoded sequence was not targeted by RNAi as confirmed by RT-PCR (Appendix A). The level of *Tb*BILBO1 remained constant. This suggests that the T2 construct is probably non-functional despite targeting correctly and is most likely degraded over time, whilst the native allele is possibly upregulated as the WB did not show a decrease in _Ty1_BHALIN levels.

### 3.11. Expression of BHALIN BILBO1-Binding Domain in BSF Does Not Rescue the RNAi-Induced Lethal Phenotype

Truncation 3 of BHALIN is 133aa longer than T2 and consists of aa 596-933, i.e., the B1B, globular and C-terminal domains (Appendix A). A viable cell-line with _cMyc_BHALIN^rec^-T3 was obtained in BSF. As with T2, _cMyc_BHALIN^rec^-T3 coding sequence did not replace a native allele but was inserted elsewhere in the genome. Immunofluorescence assays revealed that _cMyc_BHALIN^rec^-T3 was expressed and located at the FPC, but the form that it exhibited was not exactly the same as that for full-length _Ty1_BHALIN protein (Appendix A). Labelling of *T. brucei* BSF cytoskeletons showed that full-length _Ty1_BHALIN (in yellow) resembled the typical hook shape with an extension proximally towards the basal bodies. Labelling of the _cMyc_BHALIN^rec^-T3 protein (magenta) partially co-localised with _Ty1_BHALIN with strong labelling towards the basal bodies (as seen with *Tb*BILBO1) but little or no labelling on the shank (Appendix A). Similar to _cMyc_BHALIN^rec^-T2, induction of RNAi knockdown led to an unexpected rapid disappearance of the recoded _cMyc_BHALIN^rec^-T3, at 24 hpi as observed by IFA and Western blotting (Appendix A); however, the _Ty1_BHALIN protein is still present. Observation of trypanosome growth upon induction of RNAi knockdown indicated a reduction in growth rate from 24 hpi onwards, with cell death appearing at a similar rate to the _Ty1_BHALIN RNAi inducible cell line (Appendix A). Samples of mRNA were collected and RT-PCR analysis showed that mRNA from the non-recoded full-length allele of _Ty1_BHALIN was targeted by RNAi and dramatically reduced after 24 hpi, but the levels of recoded truncated allele did not reduce (Appendix A). This would suggest that the overall level of BHALIN protein is reduced (inducing the lethal phenotype at 72 hpi) and that the T3 construct, similar to T2 is non-functional despite targeting correctly. It is unclear why T3 protein is lost but one possible reason is degradation.

## 4. Discussion

In this work, we described a new protein of the hook complex that binds to BILBO1 that we name BHALIN (BILBO1 Hook Associated LINker protein). We demonstrated it to be essential for the viability of bloodstream form trypanosomes and that knockdown of BHALIN negatively influences the flagellar pocket function, resulting in a “BigEye” phenotype. *Tb*BHALIN is specific to trypanosomatids, although it is not surprising that a parasitic kinetoplastid possesses specific genes and proteins adapted to its lifestyle. In contrast, *Tb*BILBO1 has 20 orthologues (found in other *Kinetoplastida* spp.) including *Leishmania* (reviewed in [54]). This confirms the specificity of some of the FPC or HC proteins to kinetoplastids and some to specific genera within this class.

Our results confirmed the initial findings from proximity-dependent biotinylation studies suggesting a HC location of BHALIN protein in *T. b. brucei* [19] and also documented in the Tryptag database [29]. We confirmed here that in both procyclic and bloodstream forms, BHALIN is a part of the multi-protein hook-shaped complex at the flagellar pocket neck, alongside *Tb*MORN1. The hook complex is located in close proximity to the flagellar pocket collar where *Tb*FPC4 was proposed as a linker between the two structures [13]. Importantly, BHALIN also co-localises partially with *Tb*BILBO1 at the flagellar pocket collar and completely with *Tb*MORN1; it is particularly concentrated at the head of the hook that parallels the FPC. Therefore, based on its location, BHALIN could conceivably function as a linker protein between the FPC and the hook complex as previously described for the FPC4 protein. An interesting observation in BSF was that the “head” of the hook appeared to be more pointed in shape compared with the rounder “head” in PCF (Figure 2(Ga–e)). This may reflect differences in FP, FPC and HC function and structure.

Co-localisation studies in a mammalian heterologous system and in *Trypanosoma,* as well as Y2H assays confirm that BHALIN is a BILBO1 interacting partner via its *Tb*BILBO1-binding domain (B1B, T2). We previously showed that the coiled-coil region of *Tb*BILBO1 is necessary to form polymers and that the EF-hand domain plays a role in the shape of the polymers formed [14]. We demonstrate here that BHALIN binds to BILBO1 polymers and *Tb*BILBO1-T3 (aa 171-587) but not *Tb*BILBO1-T4 (aa 251-587) suggesting that the *Tb*BILBO1 EH-hand domain is required for the interaction. Two other BILBO1-binding partners interact with the EF-hand domain (the putative kinesin *Tb*927.7.3000 [14], and the newly identified *Tb*BILBO2 [55] suggesting that the *Tb*BILBO1 EF-hands play essential roles in the function of *Tb*BILBO1, such as regulation of the interaction depending on its Ca^2+^ loading status or the shape of the polymers and thus the function of the FPC.

Interestingly, deletion of the B1B domain (BHALIN-T1) was cytosolic in U-2 OS cells (even in presence of *Tb*BILBO1 polymers). However, in the parasite, a small non-cytosolic pool of BHALIN-T1 was observed along the FAZ in detergent-extracted cells. This suggests either uncomplete extraction in IFA experiments since the protein was not detected in cytoskeleton extracts by Western blotting, or that the N-terminal domain of BHALIN may be involved in an interaction with an MTQ/FAZ partner. Consequently, BHALIN could be involved in two interactions. First, an interaction with *Tb*BILBO1 at the FPC, restricting its localisation to the FPC/HC structures but also to the MTQ from the basal bodies to the FPC/HC as proven by our IFA data (Appendix A). Second, an interaction with a yet non-identified MTQ/FAZ protein that, in absence of the B1B domain, would not be restricted to the FPC/HC location. These interactions may be part of the processes involved in the building of the HC that transport protein complexes along the MTQ.

The phenotype observed after depletion of BHALIN protein in procyclic trypanosomes resembled that found by Morriswood et al. (2009) after RNAi of *Tb*MORN1 in PCF [23]. In the knockdown of both MORN or BHALIN, there was a minor delay in cytokinesis with an accumulation of trypanosomes in the 2K2N cell cycle stage. For BHALIN, there was an additional accumulation of cells in the 2K1N cell cycle stage, indicating an additional cell cycle delay. Although not lethal and representing a low proportion of the cell population, this could indicate the importance of faithful replication and division of the hook complex, which is physically connected to the axoneme. It should be noted, however, that BHALIN RNAi knockdown was unable to fully deplete BHALIN expression in PCF as ~ 26% of the protein was left and might be sufficient for cell survival, thus hiding lethal phenotypes. Indeed, RNAi silencing of *BHALIN* in bloodstream forms was lethal and protein levels were greatly reduced. This reduction in protein levels in BSF might have been the tipping-point that induced trypanosome cell death; it is undeniable that BSF tend to have stronger phenotypes from the same RNAi target of flagellum associated proteins compared to PCF [56]. The lack of lethality of RNAi BHALIN in PCF forms may reflect differences in exo- and endocytotic rates between these two life cycle stages with potentially related differences in hook complex structure and function. RNAi knockdown of BHALIN in PCF forms does not influence BILBO1 protein levels but BHALIN RNAi knockdown in BSF does appear to influence BILBO1 protein levels. With this in mind, it is possible that a combination of the reduction of both proteins may influence the phenotype produced.

A clue to the function of BHALIN can be extrapolated from observation of the phenotypes as the trypanosomes die. A “BigEye” phenotype was observed after *BHALIN* RNAi knockdown in BSF, whereby the flagellar pocket became grossly enlarged. This phenotype was first observed after RNAi knockdown of Clathrin in BSF that led to a defect in endocytosis but not exocytosis [5]. The “BigEye” phenotype was also observed in BSF after knockdown of actin or of the hook complex protein *Tb*MORN1 [25]. The fact that knockdown of HC cytoskeletal proteins leads to a defect in a membranous process was at first puzzling. One possibility might be associated with the phosphatitylinositol phosphate kinase *Tb*PIPKA that produces phosphatidylinositol 4,5 biphosphate (PI(4,5)P_2_). *Tb*PIPKA is located at the neck of the flagellar pocket in *T. brucei* and overlying the HC [57]. Interestingly, knockdown of *Tb*PIPKA is accompanied with impaired endocytosis and enlargement in the flagellar pocket that leads to a “BigEye” phenotype [57]. PI(4,5)P_2_ is crucial for endocytosis in mammalian cells and, in *T. brucei*, is concentrated at the flagellar pocket membrane [58]. If disruption of the hook complex (by depletion of BHALIN protein) influences *Tb*PIPKA function, then FP structure could be disrupted. Our functional studies using endocytic uptake assays after RNAi silencing of BHALIN in BSF *T. brucei* mimicked results obtained from *Tb*MORN1 knockdown. Our data suggest that knockdown of BHALIN is disrupting the function of the HC and leads as a consequence to the “BigEye” phenotype. Although the molecular function of BHALIN is yet to be understood, our functional studies demonstrate that BHALIN is an active part of the hook complex.

A complementation experiment using a recoded sequence was previously demonstrated with the trypanosome gene *TOEFAZ1,* rendering it resistant to RNAi [53]. The codons were changed from the native gene sequence but the amino acids coded for were retained. In our study of BHALIN, we were able to confirm that the location of the protein from a recoded copy was the same as the protein from the native allele and was fully functional as it rescued the RNAi lethal growth defect in BSF. The DNA sequences of the WT and recoded BHALIN genes are shown in Appendix A. Interestingly, the T3 truncated protein (∆Nter BHALIN) was not able to rescue the RNAi lethal growth phenotype in BSF indicating the full-length protein is required. The level of T2 and T3 truncated proteins (B1B and B1B/C-ter) were dramatically reduced as detected by IFA and WB after induction of RNAi *BHALIN*, yet we confirmed that the recoded versions of the BHALIN were not targeted by RNAi RT-PCR experiments. These two truncated proteins localised to the FPC area but did not co-localise fully with native BHALIN, indicating that although they could bind to cytoskeletal structures in the trypanosomes, they were not fully integrated into the native BHALIN structure at the HC, particularly the shank. The reduction of native BHALIN protein, most likely was sufficient to destabilise the small truncated fragments and perhaps induce their degradation. However, we cannot conclusively explain this dramatic loss of protein.

In the work presented here, we show that BHALIN is an essential protein of the hook complex in the bloodstream form of *T. b. brucei.* We were able to confirm that BHALIN and *Tb*BILBO1 interact by yeast 2-hybrid assays and when expressed together in U-2 OS cells and co-localise in vivo in trypanosomes.

A summary schematic of the location and predicted function of BHALIN in *T. brucei* is given in Figure 7 and shows that in BSF BHALIN knockdown cells the FP morphology and function are disrupted and enlarged and dextran fills the FP whilst ConA is blocked or limited in a position near the neck of the FP. This was not observed in PCF providing another example of fundamental differences in the flagellar pocket structure and function between these two proliferative forms.

## Figures and Tables

**Figure 1 microorganisms-09-02334-f001:**
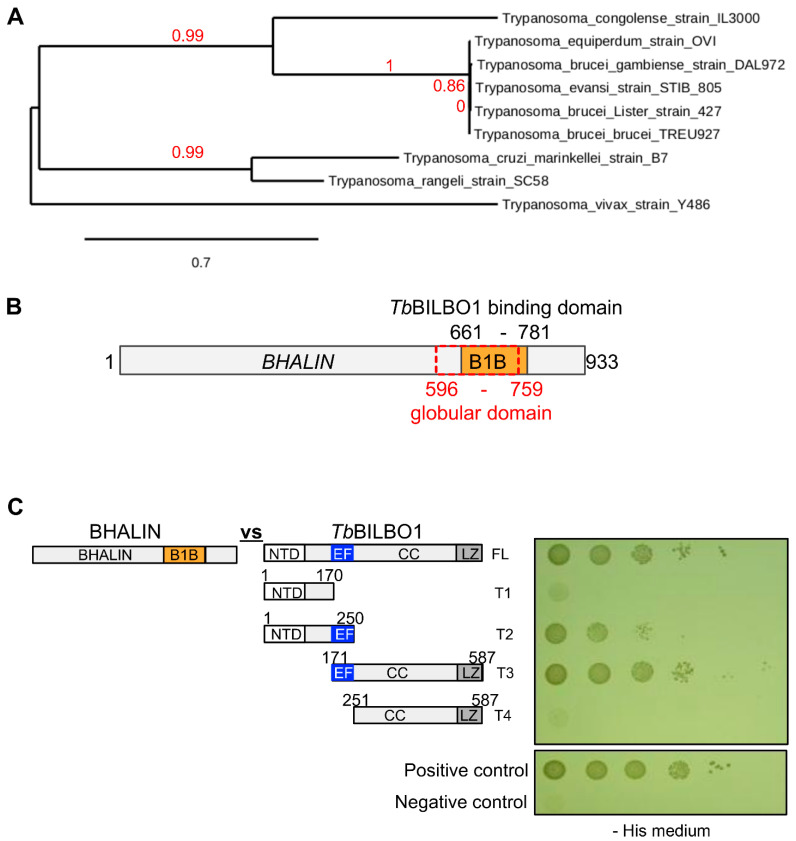
BHALIN is a conserved trypanosomatid-specific protein that interacts with *Tb*BILBO1. (**A**) Phylogenetic relationships of BHALIN orthologues in trypanosomatids, using maximum likelihood tree construction. Sequences taken from TriTrypDB and NCBI protein BLAST. Calculations made on phylogeny.fr using the ‘One Click’ mode. Branch lengths are proportional to the number of substitutions per site. Branch annotation (red numbers representing confidence estimates) are based on bootstrapping using PhyML and TreeDyn software that are automatically used in the ‘One Click’ mode. (**B**) Primary protein sequence of BHALIN showing predicted domains. B1B = BILBO1-binding domain. (**C**) Yeast-2-hybrid assay showing growth of yeast indicating interaction of BHALIN with the EF-hand domain of *Tb*BILBO1 (T2 and T3). FL = full-length, T = truncation. The interaction assay was performed on histidine depleted medium (-His).

**Figure 2 microorganisms-09-02334-f002:**
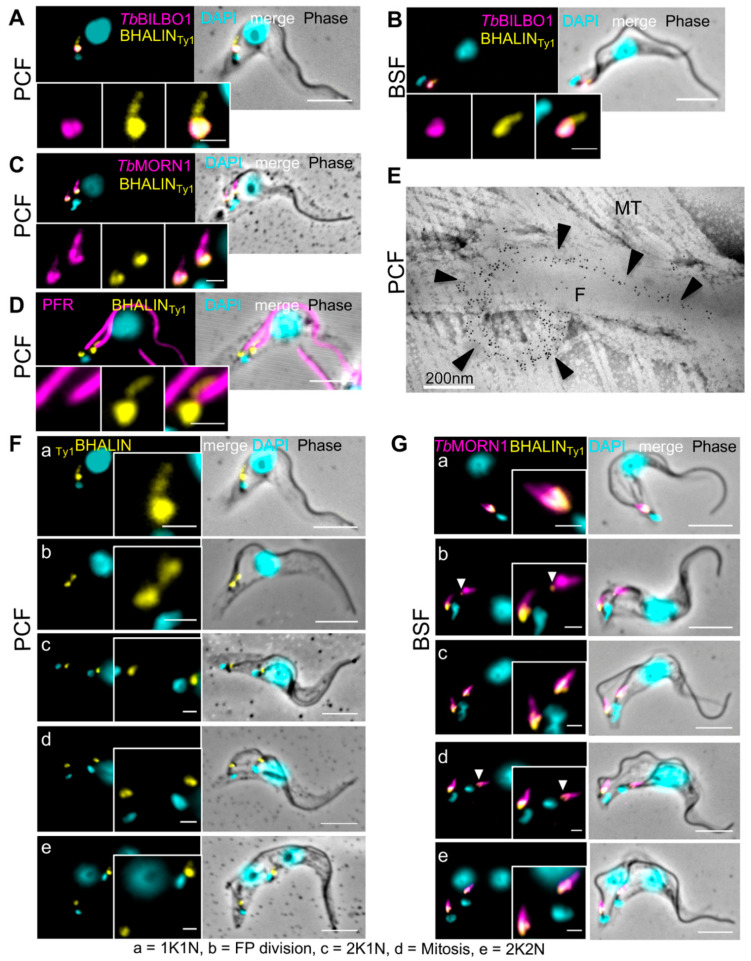
BHALIN forms a hook-shaped structure at the hook complex. Immunofluorescence assays of detergent-extracted trypanosomes. Partial co-localisation of *Tb*BILBO1 (magenta) and BHALIN (yellow) was seen in (**A**) PCF and (**B**) BSF. In both cases Ty1-tagged BHALIN exhibited a hook-shaped structure with the head at the FPC and the shank extending distally. (**C**) PCF showing co-localisation of *Tb*MORN1 (magenta) and BHALIN (yellow); the shank of the hook could be seen extending distally along the flagellum. (**D**) PCF showing co-localisation of PFR (magenta) and BHALIN (yellow); the shank of the hook is seen extending distally along the flagellum next to the PFR. (**E**) Immuno-electron micrograph of PCF flagellum with anti-Ty1 gold labelling of BHALIN; The black arrowheads indicate the regular positioning of the gold beads on the HC. F = flagellum, MT = microtubules. (**F**) PCF showing labelling of BHALIN (yellow) through the cell cycle. (**G**) BSF showing co-localisation of *Tb*MORN1 (magenta) and BHALIN (yellow) through the cell cycle. White arrowheads in G indicate the sometimes vestigial BHALIN signal on the old HC; K = kinetoplast, N = nucleus. Scale bar = 5 µm, inset = 1 µm.

**Figure 3 microorganisms-09-02334-f003:**
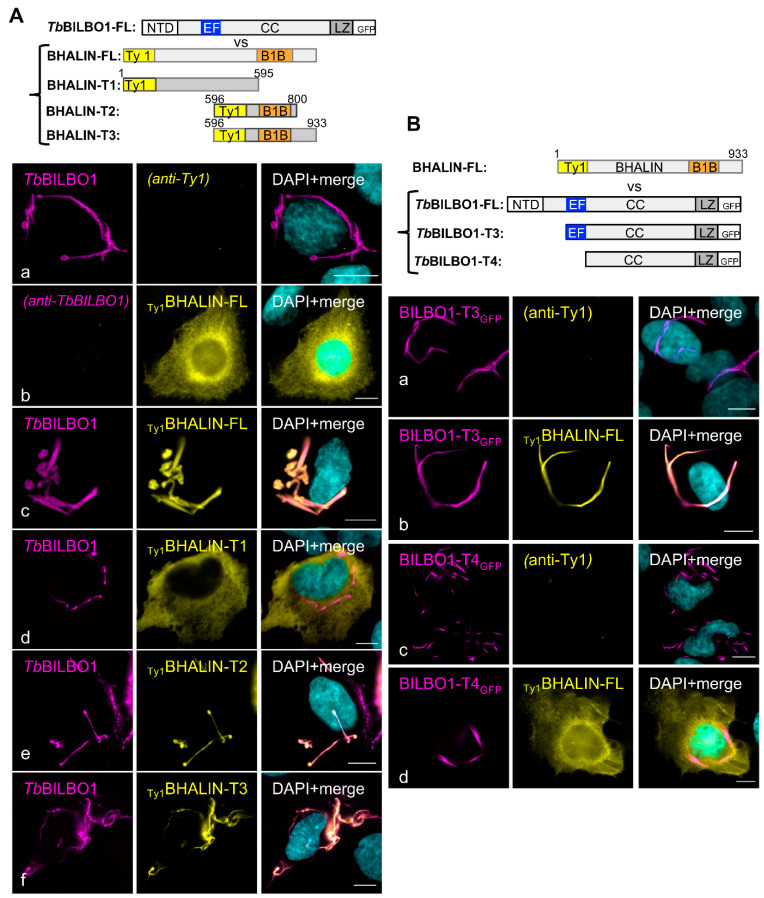
BHALIN interacts with *Tb*BILBO1 when expressed in U-2 OS cells and requires the EF-hand motifs. (**A**) Expression of full-length *Tb*BILBO1 (**a**), of _Ty1_BHALIN-FL (**b**), and co-expression of *Tb*BILBO1 and _Ty1_BHALIN-FL (**c**), _Ty1_BHALIN-T1 (**d**), _Ty1_BHALIN-T2 (**e**) and _Ty1_BHALIN-T3 (**f**); interaction was observed for BHALIN-FL, T2 and T3. (**B**) Expression of truncations of *Tb*BILBO1-T3_GFP_ and *Tb*BILBO1-T4_GFP_ without or with _Ty1_BHALIN-FL. Scale bars = 10 µm.

**Figure 4 microorganisms-09-02334-f004:**
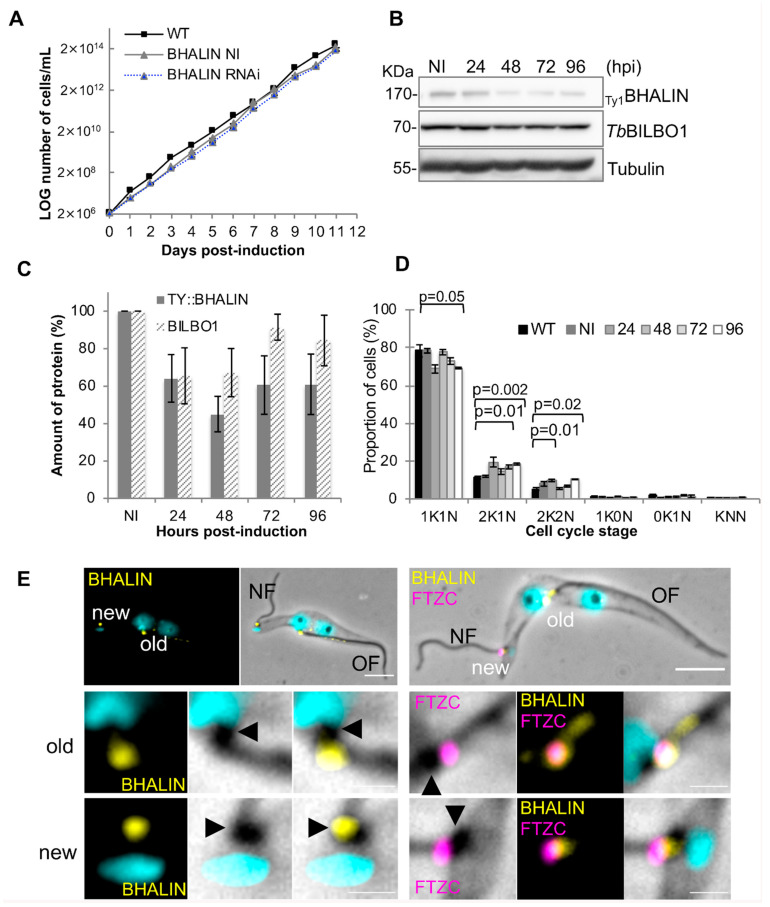
BHALIN plays a minor role in the PCF cell cycle and is associated with the basal bodies in *Tb*BILBO1 knockdown PCF cells. (**A**) Growth curve of WT PCF, non-induced (NI) and BHALIN RNAi induced trypanosomes. (**B**) Western blot of trypanosomes collected before (NI) and at 24 h intervals post-RNAi induction; tubulin used as a loading control. (**C**) Quantification of Western blots from 3 separate inductions; mean ± SE. (**D**) Bar graph to show distribution of trypanosomes in each cell cycle stage for WT, NI and at 24 h intervals post-induction; mean ±SE (student t-test). (**E**) Detergent-extracted PCF cells induced for *Tb*BILBO1 RNAi knockdown (24 H) and immunolabelled with chicken anti-BHALIN (yellow) (left panel) or with anti-BHALIN (yellow) and with anti-FTZC a transition zone marker (magenta) (right panel). The black arrowheads indicate the basal bodies. Scale bars = 5 µm, insets = 1 µm.

**Figure 5 microorganisms-09-02334-f005:**
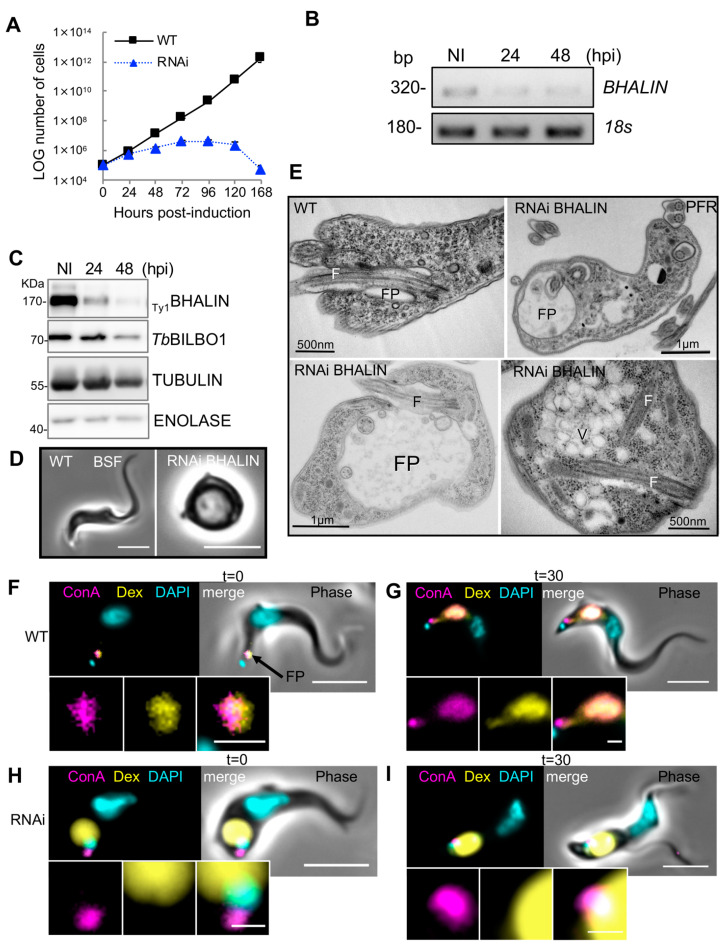
Knockdown of BHALIN in BSF leads to a “BigEye” phenotype and rapid cell death. (**A**) Growth curve of WT and BHALIN RNAi induced trypanosomes. (**B**) Semi-quantitative RT-PCR showing the decrease in BHALIN mRNA levels after RNA induction. (**C**) Western blot of NI and induced BHALIN RNAi cells over 48 h; tubulin and enolase were used as controls. (**D**) Phase contrast micrographs of BSF trypanosomes from WT cell line and 24 h post-induction of RNAi knockdown of BHALIN; scale bar = 5 µm. (**E**) Transmission electron micrographs of BSF trypanosomes from WT cell line and 24 h post-induction of RNAi induced against BHALIN knockdown. F = flagellum, FP = flagellar pocket, PFR = paraflagellar rod, V = vesicles. (**F**) A WT BSF trypanosome at time = 0 showing loading of fluorescent dyes (ConA and Dextran) in the flagellar pocket (FP). (**G**) A WT BSF trypanosome after 37 min at 37 °C showing uptake of dyes into the cell endomembrane and lysosome system. (**H**) A BSF cell induced for RNAi BHALIN knockdown (17 h) showing dextran (yellow) filling an enlarged FP and ConA (magenta) outside the FP. (**I**) A BSF trypanosomes induced for RNAi BHALIN 17 h after 30 min at 37 °C showing no entry of the dyes into the cell body. Scale bars = 5 µm, insets = 1 µm.

**Figure 6 microorganisms-09-02334-f006:**
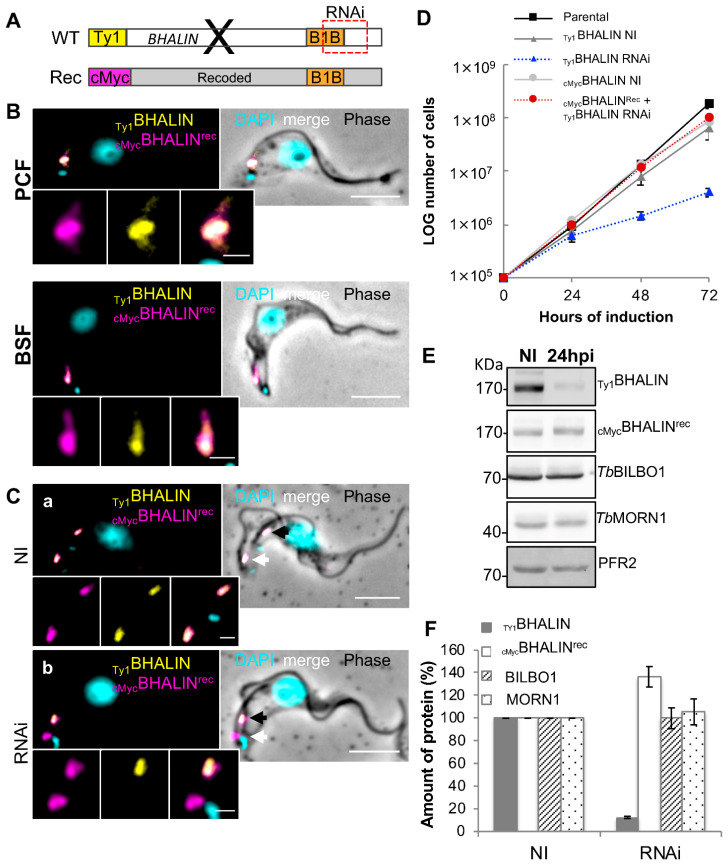
Full-length recoded BHALIN rescues the lethal RNAi phenotype in BSF. (**A**) Schematic to illustrate where *_Ty1_BHALIN* was targeted by RNAi. It also shows the recoded full-length *BHALIN*. (**B**) Immunofluorescence labelling of cytoskeletons of PCF and BSF trypanosomes expressing _Ty1_BHALIN (yellow) and _cMyc_BHALIN^rec^ protein (magenta). The recoded protein targeted to the HC and exhibited the same location and shape as _Ty1_BHALIN (yellow). (**C**) (**a**) IFA of a non-induced (NI) BSF trypanosome in the 2K1N cell cycle stage showing co-localisation of _cMyc_BHALIN^rec^ (magenta) and the _Ty1_BHALIN protein (yellow). The black arrow indicates the old hook complex and the white arrow indicates the new hook complex. (**b**) A BSF trypanosome in the 2K1N cell cycle stage after RNAi knockdown of untagged and tagged [_Ty1_BHALIN]) WT alleles for 24h, showing loss of tagged WT protein signal in the new complex but presence of the recoded _cMc_BHALIN^rec^ in both complexes. (**D**) Growth curve showing that _cMyc_BHALIN^rec^ expressing cells, (red circles), exhibited no growth defect after induction of RNAi against _Ty1_-tagged and untagged alleles. Whilst the WT cell line containing untagged and _Ty1_BHALIN (_Ty1_BHALIN-blue triangles) showed a notable growth defect from 24 hpi. Parental = the original non-modified or tagged cell line. (**E**) Western blot of RNAi resistant (_cMyc_BHALIN^rec^), (BSF trypanosomes before knockdown (NI) and 24 hpi (RNAi). _Ty1_BHALIN protein probed with anti-Ty1 and _cMyc_BHALIN^rec^ probed with anti-cMyc. *Tb*BILBO1 and *Tb*MORN1 were probed with their specific antibodies. PFR2 used as a loading control. (**F**) Quantification of Western blot from 3 separate inductions, of non-induced RNAi (NI) and 24 hpi (RNAi). Scale bars in C and D represent 5 µm, inset = 1 µm.

**Figure 7 microorganisms-09-02334-f007:**
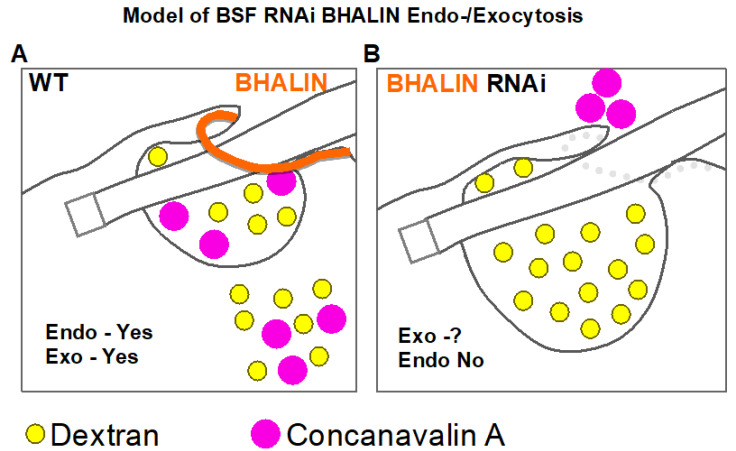
Scheme of BHALIN RNAi FP phenotype in BSF cell. (**A**) shows a scheme of a WT cell where endo- and exocytosis continues as normal when loaded with dextran (yellow circles) and ConA (larger purple circles). The orange line represents BHALIN. They enter the FP and then are endocytosed into the cell. In (**B**) a cell induced for 17 h with RNAi BHALIN is modelled and BHALIN protein is depleted (represented by grey circles). In this case, the FP is disrupted and enlarged and dextran (yellow circles) fills the FP whilst ConA is blocked or limited in a position near the neck of the FP (Adapted from Morriswood and Schmidt, 2015 [25]).

## Data Availability

The data presented in this study are available in Appendix A.

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
