# Peer review of "Bhalin, an Essential Cytoskeleton-Associated Protein of Trypanosoma brucei Linking TbBILBO1 of the Flagellar Pocket Collar with the Hook Complex"

_microorganisms, 2021, doi:10.3390/microorganisms9112334_

Round 1

Reviewer 1 Report

The manuscript by Broster-Reix investigates the cell biology of the protein BHALIN, which is a novel component of the flagellar pocket domain in T. brucei. Using a range of elegant experiments, they identify interaction partners and demonstrate a role of BHALIN in endocytosis. Of particular interest is the observation that the protein is only essential in the bloodstream form of the parasite and not in the insect, procyclic form. This affirms other data from other authors indicating fundamental differences of the endocytotic machinery between life cycle forms. This divergence is also emphasized by the observation that orthologs of BHALIN are absent in most other kinetoplastids, except T. cruzi. Overall, this is an excellent manuscript and does not require much editing. I have included some comments that the authors may or may not use for possible modifications.
Abstract
19: …a multi-role, trafficking, organelle…. Is this grammatically correct? It is an odd sentence.
Results
345: Figure legend S1 lacks a description of part D
Figure 2. What is the purpose of the black triangles within Fig. 3E? They are not mentioned in the legend.
452: How efficient was the TbBILBO1 knockdown? I can’t really see a difference in BHALIN localisation in Fig. 4E (BILBO1 knockdown) and most of the images in Fig.2F. Is there a difference?
Fig. 4B and Fig. 5C Did the home-made anti BHALIN ab not work on western blots or why was the Ty-antibody used instead?
475: I am not sure whether the BHALIN knockdown is 95% because the tubulin lane 48 hpi contains a lot less protein than the lanes NI and 24 hpi. Maybe a normalisation against tubulin would give a different value (can easily be done with ImageJ, but optional). However, the degree of knockdown is very good indeed.
500: Has the endocytosis assay also been performed with PCF cells? If so, what was the result?
Fig. S3A: The label “RNAi” has (on my copy) shifted so that the “I” is separated from “RNA”.
583: Earlier in text it was mentioned that replacement of one allele with the recoded version of BHALIN didn’t work in PCF cells. How was the truncated version integrated into the genome? Ectopically?
591: ….extremely weakly…. or …extremely weak…?
636 and Fig. S4: I am not sure whether the argument that the T2 construct is degraded over time is valid because it is constitutively expressed. Therefore, there is no timepoint zero. It should be gone at all times. This is difficult to explain. Maybe a recombination reconstituting RNAi sensitivity, although the RNA is only slightly reduced? Is Figure S4F a northern blot or a RT-PCR? Semiquantitative, endpoint RT-PCR can be very misleading? The same is true for the observations made in Fig. S5A. Unless a Northern or a proper RT-qPCR is made interpretations are difficult.
Discussion
739: The accumulation of 2K1N cells in BHALIN RNAi cells is marginal and a biological significance is disputable.
745: See comment further up. I doubt that BSF knockdown was >95%. When taking the tubulin loading into account it is probably not far off the PCF values. I am not sure whether the degree of knockdown can account for the PCF and BSF differences. Both stages seem to exhibit biological differences of the endo/exocytotic machinery. I recall that actin knockdown led to a similar phenomenon (Garcia-Salcedo, EMBO J 23, 780-789). Therefore, this observation is probably due to fundamental differences rather than experimental details.
775: Is the recoded BHALIN sequence displayed in the Supplementary data?
Figure 8: The model is very basic and not very informative. It doesn’t really say anything about function. I would not include it as a figure.
960: The Allen ref. (60) is messed up

Author Response

Reviewers comments and Authors responses Broster-Reix et. al.

Microorganisms Manuscript ID microorganisms1434145.

Submission Date 08 October 2021 Date of this review 19 Oct 2021 15:32:39

                                                                                                     26th October 2021

Dear editor,

Please find our responses to the reviewers comment for the above-mentioned manuscript.

sincerely,

Derrick R. Robinson Ph.D, FRGS. 

Director of Research (DR2/Professor)
Protist Parasite Cytoskeleton
(ProParaCyto),
 CNRS, Microbiology Fundamental and Pathogenicity, UMR 5234,
F-33000 Bordeaux, France.
Tel: + 33 55 75 74 567
Fax : + 33 557 574 803
Email - derrick-roy.robinson(at)u-bordeaux.fr
https://www.mfp.cnrs.fr/wp/la-recherche/proparacyto/
www.mfp.cnrs.fr
https://www.labex-parafrap.fr/

________________________________________________________________

Reviewer 1

Comments and Suggestions for Authors

The manuscript by Broster-Reix investigates the cell biology of the protein BHALIN, which is a novel component of the flagellar pocket domain in T. brucei. Using a range of elegant experiments, they identify interaction partners and demonstrate a role of BHALIN in endocytosis. Of particular interest is the observation that the protein is only essential in the bloodstream form of the parasite and not in the insect, procyclic form. This affirms other data from other authors indicating fundamental differences of the endocytotic machinery between life cycle forms. This divergence is also emphasized by the observation that orthologs of BHALIN are absent in most other kinetoplastids, except T. cruzi. Overall, this is an excellent manuscript and does not require much editing. I have included some comments that the authors may or may not use for possible modifications.

Abstract

19: …a multi-role, trafficking, organelle…. Is this grammatically correct? It is an odd sentence.

We have modified the sentence and it now reads as "In most trypanosomes, endo and exocytosis only occur at a unique organelle called the flagellar pocket (FP) and the flagellum exits the cell via the FP."

Results

345: Figure legend S1 lacks a description of part D

It now is described as “(D) Growth curve for WT, non-induced (-tet) and BHALIN RNAi-induced cells (+tet)“.

Figure 2. What is the purpose of the black triangles within Fig. 3E? They are not mentioned in the legend.

The triangles in Figure 2E indicate the regular positioning of the gold beads on the Hook Complex. We have now modified the legend accordingly.

452: How efficient was the TbBILBO1 knockdown?

We did not assess the efficiency of BILBO1 knockdown by western blotting in these experiments because it has already been extensively characterized and shows an optimal timing of induction of between 24 to 36 hours. This is associated with a 50% knockdown of BILBO1 protein, which is sufficient to induce the absence of biogenesis of a new flagellar pocket, a detached new flagellum followed by cell death in the majority of the population. After 48h of induction all cells are dead (Bonhivers et al. 2008, PLoS Biology).

I can’t really see a difference in BHALIN localisation in Fig. 4E (BILBO1 knockdown) and most of the images in Fig.2F. Is there a difference?

In figure 2F, the BHALIN labelling is at the FPC-HC as we initially demonstrated and in Fig. 2A, B, and C, it is a hook shape and importantly is distal to the basal bodies/kinetoplast area. Figure 4E shows that in the new flagellum (“new”), that is detached because of the BILBO1 RNAi and absence of a HC, FP and FPC, BHALIN localizes to the basal body probably via docking to an IFT or partner protein. Figure 4E has been modified to integrate an image showing co-labeling of BHALIN with anti-FTZC. FTZC is a transition zone protein and, as such, is a marker of the mature basal body transition zone region. The results and the legend were modified accordingly.

Fig. 4B and Fig. 5C Did the home-made anti BHALIN ab not work on western blots or why was the Ty-antibody used instead?

Unfortunately, the anti-BHALIN antibody has become rather weak in intensity, with considerable background and the titer had dropped considerably over time. To avoid these issues, we generated and used the endogenously tagged cell lines in most of the experiments.

475: I am not sure whether the BHALIN knockdown is 95% because the tubulin lane 48 hpi contains a lot less protein than the lanes NI and 24 hpi. Maybe a normalisation against tubulin would give a different value (can easily be done with ImageJ, but optional). However, the degree of knockdown is very good indeed.

As the reviewer mentions the “knockdown is very good indeed”. We agree, but also acknowledge that the quantification may be in error. The quantification of the blots was done with normalization of each lane against Enolase with BHALIN and BILBO1 proteins normalized proportionally. This was done on 3 separate and independent experiments. However, since there may be an error in this analysis, we have decided to remove that specific data.

500: Has the endocytosis assay also been performed with PCF cells? If so, what was the result?

No, this assay was not done on PCF for this work primarily because of the absence of any major phenotypes produced in PCF cells after BHALIN RNAi knockdown.

Fig. S3A: The label “RNAi” has (on my copy) shifted so that the “I” is separated from “RNA”.

Thank you for noticing this. It has been corrected.

583: Earlier in text it was mentioned that replacement of one allele with the recoded version of BHALIN didn’t work in PCF cells. How was the truncated version integrated into the genome? Ectopically?

This construct was successfully obtained by the replacement of one allele in PCF as an endogenous tagged protein and was checked by PCR on genomic DNA and sequencing, confirming that the recoded and tagged sequence was inserted at the right locus. We cannot explain why some allele replacements worked well whilst others failed even after numerous attempts.

591: ….extremely weakly…. or …extremely weak…?

Corrected (extremely weak)

636 and Fig. S4: I am not sure whether the argument that the T2 construct is degraded over time is valid because it is constitutively expressed. Therefore, there is no timepoint zero. It should be gone at all times. This is difficult to explain. Maybe a recombination reconstituting RNAi sensitivity, although the RNA is only slightly reduced? Is Figure S4F a northern blot or a RT-PCR? Semiquantitative, endpoint RT-PCR can be very misleading? The same is true for the observations made in Fig. S5A. Unless a Northern or a proper RT-qPCR is made interpretations are difficult.

In Figure S4F and S5F we used semi-quantitative RT-PCR to show that even if there is small decrease in RNA levels it does not explain the strong decrease in protein levels. As we had written in the manuscript, it was an unexpected finding. However, it would be surprising that, as suggested by the reviewer, recombination events occurred reconstituting the RNAi sensitivity in two different cell lines, and RNA is only slightly reduced. Furthermore, we have controlled the cell lines by PCR and DNA sequencing to confirm the presence of the recoded sequences. Finally, the full-length sequence is not affected by the RNAi because the cMycBHALINrec protein is expressed and rescues the RNAi phenotype. As the reviewer mentions the result is difficult to explain. However, we feel that one of the most likely explanations is some kind of protein degradation which is perhaps triggered by attempts to knock down the WT protein.

Discussion

739: The accumulation of 2K1N cells in BHALIN RNAi cells is marginal and a biological significance is disputable.

We thank the reviewer for this comment and we agree. To take this comment into account, we have modified the sentence that now reads as " Although not lethal and representing a low proportion of the cell population, this could indicate the importance of faithful replication and division of the hook complex, which is physically connected to the axoneme".

745: See comment further up. I doubt that BSF knockdown was >95%. When taking the tubulin loading into account it is probably not far off the PCF values. I am not sure whether the degree of knockdown can account for the PCF and BSF differences. Both stages seem to exhibit biological differences of the endo/exocytotic machinery. I recall that actin knockdown led to a similar phenomenon (Garcia-Salcedo, EMBO J 23, 780-789). Therefore, this observation is probably due to fundamental differences rather than experimental details.

We agree that both stages exhibit biological differences of the endo/exocytotic machinery. We apologize for this reference omission and have added it, we have added the following to the text: "Interestingly, a differential role for actin during the life cycle of T. brucei was observed. RNAi knockdown of actin in PCF had no effect whilst it was lethal in BSF leading to a similar BigEye phenotype [9]". We have already responded to the knockdown issue and removed the quantification data.

775: Is the recoded BHALIN sequence displayed in the Supplementary data?

We have included a file (Figure S6) that indicates that there is 73% nucleotide identity between the native WT and recoded BHALIN sequences. However, this difference was sufficient to allow the recoded BHALIN to be resistant to RNAi knockdown.

Figure 8: The model is very basic and not very informative. It doesn’t really say anything about function. I would not include it as a figure.

We agree that the model is very basic and does not address function per se, but we feel that it provides a rapid explanation of the phenotype obtained after BSF BHALIN RNAi knockdown, and as such, provides a notable take-home message. For these reasons (and with respect to the reviewer) we would like to keep this image.

960: The Allen ref. (60) is messed up

Thank you for noticing this. The error has been fixed.

Reviewer 2 Report

The authors described the functional characterization of the BHALIN – (BILBO1 Hook Associated Linker) a conserved trypanosomatid-specific protein that interacts with TbBILBO1 as demonstrated by yeast 2-hybrid system and is in the vicinity of TbMORN1 as demonstrated by proximity-dependent biotin identification protein.

The main results showed in this manuscript are:

  1. Using yeast two-hybrid assays it was shown that BHALIN interacts with full-length TbBILBO1 and this interaction occurs through the TbBILBO1 EF-hands domain.
  2. Localization of the protein: using tagged BHALIN with the 10xTy1 tag on either the N- or C-terminal region and IF microscopy it was shown that BHALIN localizes at the Flagellar Pocket Collar (FPC)/hook complex region through the cell cycle, originating a hook-shaped structure that it is cytoskeleton-associated. BHALIN partially colocalizes with BILBO1 at the Flagellar Pocket Collar and with MORN1 (marker for the hook complex) either in Blood stream forms or in procycling forms (PCF) of Trypanosome. Using electron microscopy, it was confirmed the localization at FPC and also the flagellum co-localizing with Morn1 which supports the detection of BHALIN as a member of the TbMORN1 proximity-dependent biotin network.
  3. Functional/Interacting domains: In mammalian U-2 OS cells BHALIN is cytoplasmic and colocalizes with the filaments formed by BILBO when both are co-expressed, supporting their interaction in a heterologous system. BHALIN seems to affect the BILBO filaments (bundling???). Using truncated versions of BHALIN and BILBO the authors showed that BHALIN interacts with BILBO through the half C-terminal region and confirmed the requirement of the EF-hand-domain of BILBO for the interaction.
  4. BHALIN Knockdown in PCF an BSF: In PCF BHALIN KD affects trypanosome cell cycle and looses its localization. In contrary Knockdown of BHALIN in bloodstream forms (BSF) is lethal and induces a "BigEye" phenotype. BILBO KD affects BHALIN localization that is observed at the FPC region of the old flagellum but localized at the mature basal body in the new flagellum indicated mislocalization of BHALIN in BILBO low levels background.
  1. BHALIN KD phenotypes rescue: RNAi-resistant BHALIN rescues the lethal phenotype in BSF and the cell cycle phenotypes in PCF. BHALIN BILBO1-binding domain in BSF does not rescue the RNAi-induced lethal phenotype
  2. In Trypanosoma cells (only PCF) truncated forms of BILBO (Deletion of the BILBO1-binding and C-terminal domains) affects BHALIN localization supporting the observations in the heterologous localization in mammalian cells suggesting that the B1B domain in C-terminal half of the protein is required for its localization at the Flagellar Pocket Collar/hook complex region.

The results are clear and the rescue of the BHALIN knockdown phenotypes are convincing and therefore the manuscript deserves to be published. However, before publication this referee still has some concerns that should be approached.

  1. The authors mentioned that (line 382) “ As the new flagellum grew and a new FP was formed, the BHALIN signal could be observed as two structures of equivalent size and intensity (Figures 2F, b, c). As the kinetoplast divided, followed by mitosis (Figures 2F and G, d), the signal for BHALIN remained present at both hook complexes (Figure 2F and G, e). A close analysis of the staining in the BS forms show that BHALIN is sometimes vestigial in one of the structures stained by tbMORN1 as is the case of Fig. 2G b and d. This is not mentioned in the text or commented. This should be clarified, and arrows would be welcome in this figure.
  2. In figure 4E a basal body marker should be used to unequivocally show what the authors are reporting.
  3. The knockdown of BHALIN in bloodstream forms is lethal and induces a "BigEye" phenotype but also causes the decrease of the amount of BILBO1 protein (see Figure 5C and D). This result is not mentioned in the text and is quite important because the lethality and the "BigEye" phenotype could be a combined result due to low levels of BHALIN+BILBO1. The decrease of BILBO levels in the knockdown background is not observed in PCF (see figure 4) which could justify the differences. This requires clarification.
  4. Finally, the organization of the manuscript is not always fluid. For example, “protein domains” vs interaction and localization, using truncated forms of BHALIN and BILBO1 in mammalian cells are in Figure 3 and then we come back to these issues using Trypanosomes cells in figure S4 and S5. These data could be integrated together allowing for a joined discussion. I am not sure if the Trypanosome lines expressing truncated version of BHALIN should be supplementary figures. Maybe it is possible to combine part of Fig. 4 with these supplementary figures…

Author Response

Reviewer 2

The authors described the functional characterization of the BHALIN – (BILBO1 Hook Associated Linker) a conserved trypanosomatid-specific protein that interacts with TbBILBO1 as demonstrated by yeast 2-hybrid system and is in the vicinity of TbMORN1 as demonstrated by proximity-dependent biotin identification protein.

The main results showed in this manuscript are:

  1. Using yeast two-hybrid assays it was shown that BHALIN interacts with full-length TbBILBO1 and this interaction occurs through the TbBILBO1 EF-hands domain.
  2. Localization of the protein: using tagged BHALIN with the 10xTy1 tag on either the N- or C-terminal region and IF microscopy it was shown that BHALIN localizes at the Flagellar Pocket Collar (FPC)/hook complex region through the cell cycle, originating a hook-shaped structure that it is cytoskeleton-associated. BHALIN partially colocalizes with BILBO1 at the Flagellar Pocket Collar and with MORN1 (marker for the hook complex) either in Blood stream forms or in procycling forms (PCF) of Trypanosome. Using electron microscopy, it was confirmed the localization at FPC and also the flagellum co-localizing with Morn1 which supports the detection of BHALIN as a member of the TbMORN1 proximity-dependent biotin network.
  3. Functional/Interacting domains: In mammalian U-2 OS cells BHALIN is cytoplasmic and colocalizes with the filaments formed by BILBO when both are co-expressed, supporting their interaction in a heterologous BHALIN seems to affect the BILBO filaments (bundling???). Using truncated versions of BHALIN and BILBO the authors showed that BHALIN interacts with BILBO through the half C-terminal region and confirmed the requirement of the EF-hand-domain of BILBO for the interaction.
  4. BHALIN Knockdown in PCF an BSF: In PCF BHALIN KD affects trypanosome cell cycle and looses its In contrary Knockdown of BHALIN in bloodstream forms (BSF) is lethal and induces a "BigEye" phenotype. BILBO KD affects BHALIN localization that is observed at the FPC region of the old flagellum but localized at the mature basal body in the new flagellum indicated mislocalization of BHALIN in BILBO low levels background.
  5. BHALIN KD phenotypes rescue: RNAi-resistant BHALIN rescues the lethal phenotype in BSF and the cell cycle phenotypes in BHALIN BILBO1-binding domain in BSF does not rescue the RNAi-induced lethal phenotype.
  1. In Trypanosoma cells (only PCF) truncated forms of BILBO (Deletion of the BILBO1-binding and C-terminal domains) affects BHALIN localization supporting the observations in the heterologous localization in mammalian cells suggesting that the B1B domain in C-terminal half of the protein is required for its localization at the Flagellar Pocket Collar/hook complex region.

The results are clear and the rescue of the BHALIN knockdown phenotypes are convincing and therefore the manuscript deserves to be published. However, before publication this referee still has some concerns that should be approached.

  1. The authors mentioned that (line 382) “ As the new flagellum grew and a new FP was formed, the BHALIN signal could be observed as two structures of equivalent size and intensity (Figures 2F, b, c). As the kinetoplast divided, followed by mitosis (Figures 2F and G, d), the signal for BHALIN remained present at both hook complexes (Figure 2F and G, e). A close analysis of the staining in the BS forms show that BHALIN is sometimes vestigial in one of the structures stained by tbMORN1 as is the case of 2G b and d. This is not mentioned in the text or commented. This should be clarified, and arrows would be welcome in this figure.

We thank the reviewer for this comment. We agree that the signal seems weaker, and have accordingly modified the text and figure legends. The text now includes the sentence; “A close analysis of the immunolabelling in the BS forms show that BHALIN signal is sometimes vestigial on the old HC, (Figures 2Gb and d).” We have also added white arrowheads to indicate this.

  1. In figure 4E a basal body marker should be used to unequivocally show what the authors are reporting.

We have modified the figure by adding a co-labelling with an anti-FTZC. FTZC is a marker of the transition zone of the mature basal body. We have modified the results and the figure legend accordingly.

  1. The knockdown of BHALIN in bloodstream forms is lethal and induces a "BigEye" phenotype but also causes the decrease of the amount of BILBO1 protein (see Figure 5C and D). This result is not mentioned in the text and is quite important because the lethality and the "BigEye" phenotype could be a combined result due to low levels of BHALIN+BILBO1. The decrease of BILBO levels in the knockdown background is not observed in PCF (see figure 4) which could justify the This requires clarification.

We agree with the reviewer that RNAi knockdown of BHALIN in PCF does not influence BILBO1 levels but BHALIN RNAi knockdown in BSF could influence BILBO1 protein levels, and that this may influence the phenotype produced. We have clarified this point and now mention in the discussion that : “RNAi knockdown of BHALIN in PCF forms does not influence BILBO1 protein levels but BHALIN RNAi knockdown in BSF does appear to influence BILBO1 protein levels. With this in mind, it is possible that a combination of the reduction of both proteins may influence the phenotype produced”.

  1. Finally, the organization of the manuscript is not always fluid. For example, “protein domains” vs interaction and localization, using truncated forms of BHALIN and BILBO1 in mammalian cells are in Figure 3 and then we come back to these issues using Trypanosomes cells in figure S4 and These data could be integrated together allowing for a joined discussion. I am not sure if the Trypanosome lines expressing truncated version of BHALIN should be supplementary figures. Maybe it is possible to combine part of Fig. 4 with these supplementary figures…

We respectfully disagree with the reviewer about the organization of the manuscript.  Although we agree that there are numerous ways to do this, we feel that in its current state it is a good way to explain a complex set of experiments. We feel that the section based on, “protein domains” vs interaction etc, in mammalian cells in Figure 3”, is a strong follow-up on the previous point that BHALIN is located near to the FPC/HC, and this is done by showing that BILBO1 and BHALIN interact in a heterologous system. We didn’t want to flip from a mix of mammalian and parasite truncations, then RNAi, then back to parasite truncations, and used the mammalian truncation work to simply show the usefulness of the interaction system in a non-parasite environment. It also clarifies, early on, which BHALIN domain interacts with which BILBO1 domain.  Finally, given the short response time assigned by the journal for the authors to respond, (5 day extended to 9), we don’t feel that a major re-organization would help the manuscript.